# Sentinel-1 cross-polarization ratio as a proxy for surface mass balance across east Antarctic ice rises

Thore Kausch[1], Stef Lhermitte[1,4], Marie G.P. Cavitte[2], Eric Keenan[3], and Shashwat Shukla[1]

[1]Delft University of Technology, Mekelweg 5, 2628 CD Delft, The Netherlands
[2]Earth and Life Institute, Université catholique de Louvain, Louvain-la-Neuve, Belgium
[3]University of Colorado Boulder, Department of Atmospheric and Oceanic Sciences, 4001 Discovery Dr., CO 80309 Boulder, The United States of America
[4]Department of Earth and Environmental Sciences, KU Leuven, Celestijnenlaan 200E, 3001 Leuven, Belgium

**Correspondence:** Thore Kausch (t.kausch@tudelft.nl)

**Abstract.**

The determination of the Surface Mass Balance (SMB) for the Antarctic ice sheet remains subject to significant uncertainty. Sentinel-1 Synthetic Aperture Radar (SAR) satellite sensors with their large spatial coverage and ability to penetrate the snowpack, represent a promising tool to more effectively assess the SMB. However, it is challenging to directly relate SMB to the SAR backscatter signal. The multitude of interactions between the snow microstructure and the backscatter signal complicate a direct translation from the backscatter signal to SMB using physical models. Additionally, the lack of reliable ground truth data limits the establishment of an empirical relationship with SMB across all of Antarctica. In this study we focus on establishing an empirical relationship between the SMB and dual polarisation SAR backscatter locally across three ice rises in Dronning Maud Land. The SMB of the ice rises was reconstructed using ground penetrating radar data and compared to the incidence angle corrected, four year average of the Sentinel-1 cross-polarization ratio $\sigma_{hh}$ / $\sigma_{HV}$. We found a correlation between the SMB and the cross-polarization ratio with an R-value of 0.65 when using all available orbits. To understand this relationship we ran a radiative transfer model (SMRT) together with a physical snowmodel (SNOWPACK), which was forced by field measurements across the central ice rise. The results show generally lower density and optically equivalent grain diameter in accumulation zones but also higher specific surface area of the grains. Overall the results show the existence of a relationship between the SMB and the cross-polarization ratio for the study area. This promising proxy could be combined with physical models and extended to larger parts of Antarctica in future research.

*Copyright statement.* TEXT

## 1 Introduction

Measuring Antarctica's mass balance is key to assessing its contribution to sea level rise. Yet estimating the absolute mass balance of Antarctica, and especially East Antarctica, remains challenging and is only possible with high uncertainties (Lenaerts

et al., 2019; The IMBIE team, 2018; Rignot et al., 2019). For example, The IMBIE team (2018) found a mass balance for East Antarctica between 1992 - 2017 is $5 \pm 46 \ \mathrm{Gt/year}$. One reason for these high uncertainties are difficulties in quantifying the surface mass balance (SMB), which is defined as the annual sum of all surface processes (e.g. snowfall, sublimation, runoff, wind erosion/deposition) that affect the net accumulation or erosion at the surface of an ice sheet.

5    Currently, SMB is usually quantified by regional climate models like RACMO2 (van Wessem et al., 2018), MAR (Agosta et al., 2019) or COSMO-CLM (Souverijns et al., 2019), as in-situ measurements are sparse and do not cover the whole Antarctic continent. The in-situ measurements are however crucial to evaluate climate models. The most common in-situ measurements are automatic weather stations (AWS), snow stakes or firn cores. Alternatively, SMB can be assessed across tracks in the field by combining ground penetrating radar (GPR) measurements with ice core dating. This method allows for the reconstruction 10 of the SMB over the last decades and adds spatial coverage to the data (Drews et al., 2015; Kausch et al., 2020; Cavitte et al., 2022). The drawback of this GPR method, however, is the need for an ice core, which is costly and time consuming. Additionally the spatial coverage of this method is limited by the slow driving speed needed to record the GPR data. More ground can be covered using airborne radar, which however is expensive and has large uncertainties due to the unknown density of the snow, as well as differences in density between dry snow, wet snow, firn and ice (Medley et al., 2013).

15    Satellite remote sensing presents an attractive alternative way of quantifying SMB, because of its frequent revisit times (up to weekly and daily) and higher spatial coverage with a spatial resolution of 20 x 40 m (Sentinel-1 Extra Wide swath) and up to 2 x 1 m in certain cases (TerraSAR-X Spotlight mode). Synthetic Aperture Radar (SAR) is especially attractive as its signal is sensitive to snow micro-structure (Mätzler, 1998; Joughin et al., 2016; Lievens et al., 2019), independent of sunlight and less restricted by cloud cover. The SAR signal penetrates several meters into the snow and interacts with the surface layer, internal 20 layers as well as individual grains within the snowpack. As a result of these multiple interactions, it is difficult to derive SMB directly from SAR-derived observations. Nevertheless, an increasing amount of successful applications have been developed that derive snow parameters from satellite radar data: e.g. retrieving snow depth based on the co-polar phase differences (CPD) of TerraSAR-X acquisitions (Patil et al., 2020) or relating interferometric radar signals to physical snow properties (Rott et al., 2021). Recently CPDs have also been used to successfully derive firn thickness values in western Greenland (Parrella et al., 25 2021). One particularly promising approach used the cross-polarization ratio $\sigma_{VH}/\sigma_{VV}$ of Sentinel-1 to map snow depth on the whole northern hemisphere (Lievens et al., 2019). This approach builds on the assumption that the cross-polarization ratio increases with snow height as a result of the longer travel path of the radar signals through the snow column. The longer travel path allows for more multiple and anisotropic scattering, which increases the cross-polarized part of the signal ($\sigma_{VH}$). Although this approach is promising, it can not be directly translated to Antarctica, as it depends on the radar signal penetrating 30 through the whole snowpack and returning from the ground and requires $\sigma_{VH}/\sigma_{VV}$ data which are not widely available over Antarctica. Lievens et al. (2019), however, highlights that the radar wave interactions within the snowpack and the volume scattering response of the snowpack are what is driving cross-polarization ratio variability. Furthermore, it highlights that important variables such as snow height can be derived from these interactions.

    In this study, we apply a similar approach by correlating the Sentinel-1 cross-polarization ratio $\sigma_{HV}/\sigma_{HH}$ (instead of 35 $\sigma_{VH}/\sigma_{VV}$) to the SMB of three Antarctic ice rises. Although $\sigma_{HH}$ is generally less sensitive to volume scattering within the

snowpack compared to $\sigma_{VV}$, studies have demonstrated a relationship between $\sigma_{HH}$ and snow water equivalent (SWE) (Ulaby and Stiles, 1980), as well as a notable increase in cross-polarization ratio with snow depth (Stiles and Ulaby, 1980). In addition to assessing the correlation, we aim to understand the physical processes driving the cross-polarization ratio variability, which relates to volume scattering from the snowpack. Detailed information of the snow microstructure is required to understand

the volume scattering response of a snowpack. Therefore the goal of this study is twofold. The first goal of this paper is to utilize field measurements across three East Antarctic ice rises to assess the empirical relationship between their SMB and the Sentinel-1 signal. The second goal is to examine spatial and temporal snow microstructure patterns across the central ice rise in order to explain the empirical relationship. To achieve the first goal we use the reconstructed SMB from a set of 20 ground-penetrating radar tracks across the three ice rises (Cavitte et al., 2022, Fig. 1). For the second goal, we utilize the Snow

Microwave Radiative Transfer model (SMRT; Picard et al., 2018a, b) to derive the theoretical backscatter of the snowpack at one of the ice rises. By comparing the theoretical backscatter intensity to the observed backscatter intensity of Sentinel-1, we are able to investigate to what degree snowpack parameters such as optically equivalent grain diameter and density can explain the SMB-cross-polarization relationship. Since detailed information about the snow microstructure are required to run SMRT, we use the physics-based snow model SNOWPACK (Bartelt and Lehning, 2002; Lehning et al., 2002a, b), forced by two years

of AWS measurements, to model the temporal evolution of snow parameters such as density and optically equivalent grain diameter as well as their spatial variability across the ice rise.

## 2 Data and study area

### 2.1 Study area

The field measurements were taken on three ice rises (the Lokkeryggen ice rise (LIR), the Hammarryggen ice rise (HIR) and

the Derwael ice rise (DIR); Fig. 1) in Dronning Maud Land in coastal East Antarctica. Ice rises represent an elevated part of the ice shelf where the ice rests on top of local island-like topography, instead of the ocean (Matsuoka et al., 2015). Some ice rises are peninsula-like and extend out from the Antarctic continent, and are therefore partially surrounded by the ice shelf. Because of their relative topography, compared to the surrounding ice shelf, ice rises have a large impact on the SMB of the ice shelf as their windward flanks usually represent high accumulation areas. This high accumulation is caused by the combination of

katabatic winds and orographic uplift of moist air on the upwind flank of an ice rise (Lenaerts et al., 2014; Kausch et al., 2020). In addition, ice rises are natural pinning points of the ice shelf resulting in very low ice flow speeds at the center of the ice rise (Matsuoka et al., 2015). These two characteristics make ice rises an ideal location to investigate the influence of the SMB on Sentinel-1 backscatter. Firstly, because the high accumulation on the windward flank of the ice rise results in large SMB variability over a short distance (Kausch et al., 2020), this allows for for a sensitivity study of backscatter intensity to changes

in the SMB. Secondly, because areas with low ice flow speeds are less susceptible to fracturing and crevassing which creates a strong surface scattering response, the backscatter signal from the undisturbed snowpack will dominate. Lastly, the higher relative altitude of the ice rise reduces the occurrence of melt.

The LIR is ~ 350 m high and ~ 25 km wide in east west direction. It is oriented from the north to the south, being surrounded by the ice shelf from the east, north and west and connected to the grounded ice sheet towards the south. The LIR has a local ice divide oriented north-south with gentle slopes below 1 °moving sideways east or west of the divide. The HIR is located ~ 50 km to the west of the LIR and is ~ 360 m high, ~ 50 km wide and represents a triple junction for the ice flow (Fig. 1). Similar to LIR, HIR is connected to the main ice sheet in the south with slopes below 1 °, except at the grounding line where slopes can reach 2 °. Finally, DIR, which is 40 km long and ~ 38 km wide, is the highest of the three ice rises with a maximum height of ~ 450 m and with no land connection to the grounded East Antarctic ice sheet.

The whole study area is characterized by a combination of steady katabatic winds from the south east and synoptic air masses originating the north east (Lenaerts et al., 2014). As a result, the two AWS's measure an average southeast wind direction of ~ 132 °, which leaves the eastern flanks of the ice rises as the windward side and the western flanks of the ice rises as the leeward side.

## 2.2   Automatic weather stations

Two automatic weather stations (AWS) were deployed on the leeward (AWS1) and windward (AWS2) flanks of the LIR (Fig. 1) between 11 Dec 2017 and 6 May 2018 for AWS1 and between 11 Dec 2017 and 12 Dec 2019 for AWS2. Both AWS's measured 2 meter air temperature, relative humidity, wind speed and direction, incoming and outgoing shortwave and long wave radiation at the surface, as well as snow depth with a temporal resolution of 10 min. Since the wind sensor of AWS 2 failed to record any data between August and December 2018, we used ERA5 reananalysis data (Hersbach et al., 2020) to interpolate this gap in the wind data of AWS 2. The ERA5 wind data has a pixel size of 28 km and a bias of -0.45 m/s for the time periods where AWS data and ERA5 data are available, which is from January to July in 2018 and from January to October in 2019.

## 2.3   Ground penetrating radar

GPR tracks were recorded on each of the three ice rises (Fig.7). During the 2017 Mass2Ant field campaign, several GPR tracks were recorded on LIR using a 400 MHz antenna (GSSI:SIR 3000). This includes a ~20 km east to west GPR profile across LIR. This set up was repeated during the 2018 Mass2Ant campaign over the HIR, where three ~10 km profiles were recorded across the three ice divides of the triangular ice rise (Cavitte et al., 2022). The GPR data was dated at ice cores, a full analysis of the ice cores can be found at Philippe et al. (2016) and Wauthy et al. (2024). This dated GPR data was used to reconstruct the SMB across LIR, a description of this can be found in the methods section (3.2). Additionally, we use GPR tracks from the DIR, which were collected in 2012 using the same set up (Drews et al., 2015). In all cases, the antenna was dragged behind a skidoo to survey the top 50 m of the snowpack.

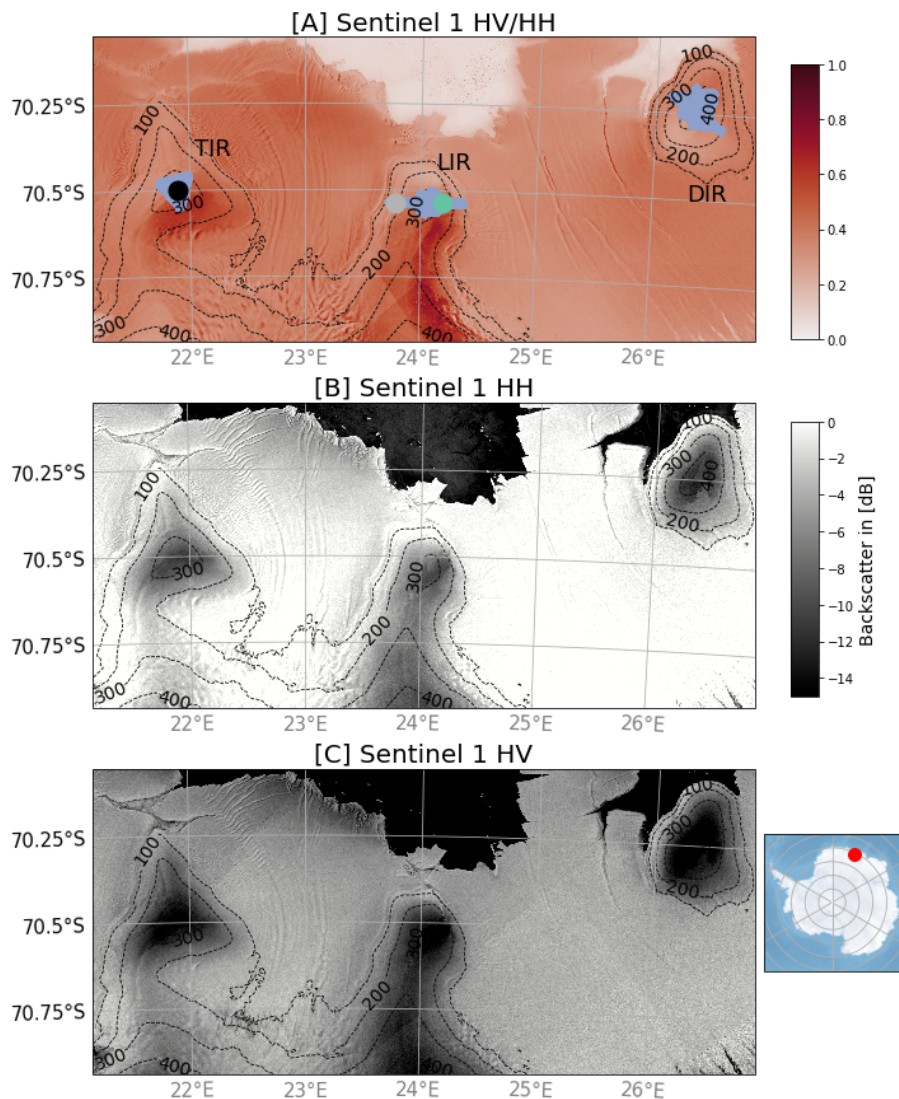

**Figure 1.** Overview of the study area. A) shows the cross-polarisation ratio ($\sigma_{HV}/\sigma_{HH}$) from Sentinel-1 data as well as the location of the two AWS (green points), the GPR track (blue line) and the location of the TLS (black point). B) and C) show the $\sigma_{HH}$ polarization and the $\sigma_{HV}$ polarization respectively. The Backscatter in [dB] colorbar applies to B) and C). The red dot on the Antarctica map locates panels A)-C).

## 2.4 Snow micro pen

During the 2019 field campaign, a SnowMicroPen (SMP) was employed at LIR to measure density and optically equivalent grain diameter within the first 1.25 m of the snowpack. The SMP works by driving a metal pole into the snowpack at a fixed speed and precisely measuring the force required to penetrate the snowpack with a vertical resolution of 1.25 mm. This force

can be translated to density, correlation length, and specific surface area (SSA) using an empirical approach developed by Proksch et al. (2015). Here used the improved empirical relationship by Calonne et al. (2020) to obtain density, correlation length and SSA from the SMP measurements and in a second processing step derived the optically equivalent grain diameter from the SSA using the geometrical relationship:

$$GS = \frac{6}{(SSA * \rho_{ice})} \tag{1}$$

Where GS is optically equivalent grain diameter in $m$, SSA is the specific surface area in $m^2/kg$ and $\rho_{ice}$ is the density in $kg/m^3$. Equation 1 follows simple geometry rules but assumes spherical grains. This is important to note, as high SSA is a sign of a non-spherical grain shape. Therefore the optically equivalent grain diameter calculated here should be seen as an approximation and not a direct measurement. Especially in areas of high SSA, real optically equivalent grain diameter might vary from the SMP calculations.

To accurately cover the high spatial variability within snowpack parameters, especially near the surface, we conducted a total of 275 SMP measurements across LIR. These 275 measurements are distributed into 25 measurement locations along the GPR track (Fig. 5). At each of these locations, 11 measurements were taken along a 20 m track perpendicular to the main track, with a SMP measurement every 2 m. These 11 measurements at each station were averaged into one depth profile per location.

## 2.5 Sentinel-1

A Copernicus Sentinel-1 image collection of $\sigma_{HH}$ and $\sigma_{HV}$ polarisation imagery was created over the study area consisting of 363 ascending C-Band Sentinel-1 SAR images in extra wide (EW) mode from relative orbits 30, 59, 88, 132 and 161 between 2014 and 2020. These images are evenly distributed across all months, with the highest count of 38 in October, November, and December and the lowest count of 24 in April, and an average of 32 images a month. All five orbits cover LIR, whereas all but orbits 88 and 30 cover HIR/DIR, respectively. The image collection was created using Google Earth Engine (GEE) (Gorelick et al., 2017) using the default Sentinel-1 pre-processing steps including thermal noise removal, radiometric calibration and terrain correction. Based on the available dual-polarization imagery, we calculated for each image the cross-polarization ratio defined as the logarithmic ratio between $\sigma_{HV}$ and $\sigma_{HH}$ to create cross-polarization ratio time series. For each orbit, we calculated the time-averaged cross-polarization ratio between 2014 and 2020. To compare the Sentinel-1 data to the GPR SMB , we resampled the cross-polarization ratio at the SMB measurement locations to a 50 m resolution using spatial averaging. To investigate the temporal variability of the Sentinel-1 data we also created time series (Fig. 8) based on the spatial mean backscatter intensity within a 200 x 200 m around the location of AWS2. The satellite sampling area is larger, compared to the SMP sampling area to allow for spatial averaging which is necessary to avoid speckle.

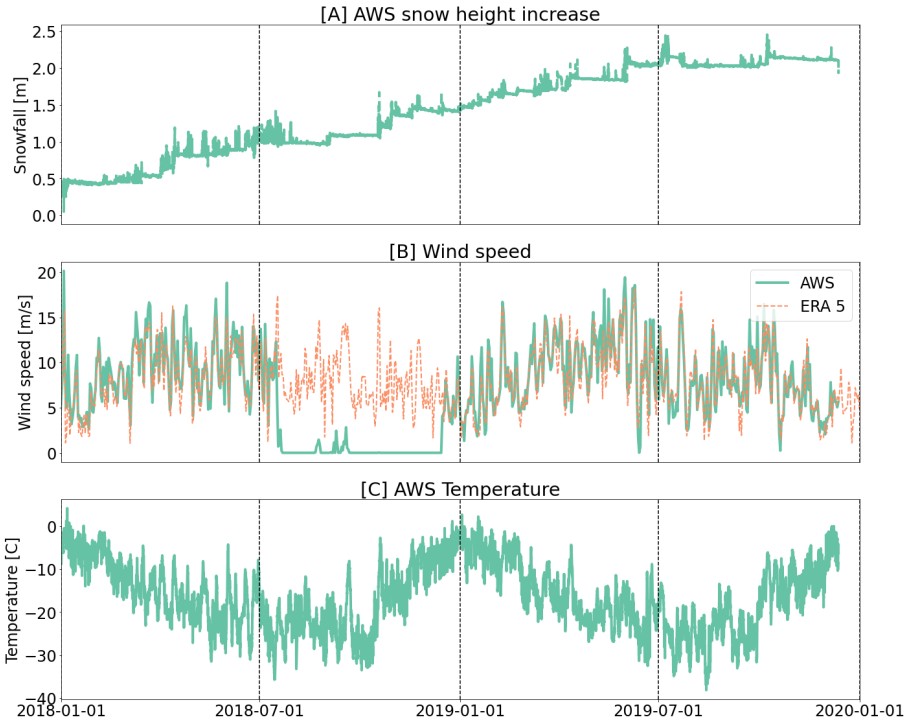

**Figure 2.** Overview of field measurements. A) shows the snow height measured by AWS2 from 2018 to 2020. B) shows the wind speed measured by AWS2 as well as the wind speed from ERA5, which was used to fill the gap in late 2018. C) shows the temperature measured by AWS2.

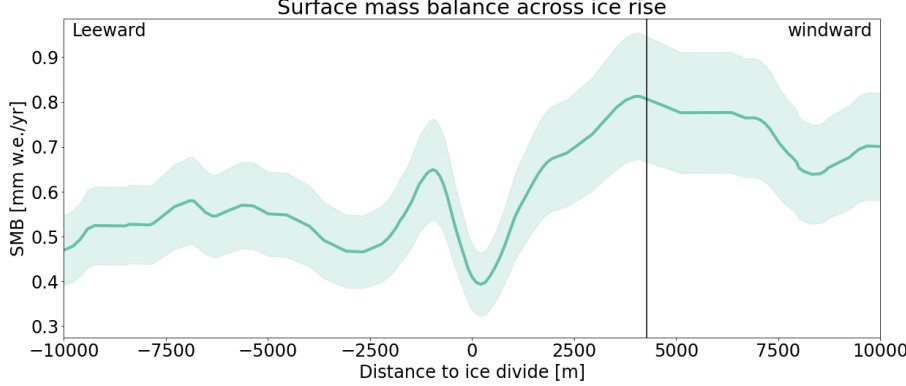

**Figure 3.** Surface mass balance as it was reconstruted from the GPR data across LIR. The location of the GPR track is shown in Fig. 1. The black horizontal line marks the location of AWS2.

## 3 Method

### 3.1 Overview

In this study we applied four main processing steps to analyse the relationship between the SMB and the Sentinel-1 cross-polarisation ratio. The first step is to reconstruct the SMB across the three ice rises by combining the ground penetrating radar data with the ice core dating. The second step is the processing of the Sentinel-1 data, where we stacked the Sentinel-1 cross-polarization ratio of all available orbits for a time period of 6 years while correcting for the effect of the incidence angle. Additionally for step three and four, we focused our analysis on the LIR, where we combined the physical snowmodel SNOWPACK with the radiative transfer model SMRT to create a synthetic backscatter signal at the location of AWS2. By comparing the synthetic backscatter of SMRT with the observed backscatter, we are able to investigate to what degree the observed backscatter variability can be explained by snow microstructure properties such as optically equivalent grain diameter and density.

### 3.2 SMB from ground penetrating radar

The average SMB was reconstructed from the ground-penetrating radar (GPR) measurements and the ice cores across the three ice rises using a method similar to Kausch et al. (2020) and Cavitte et al. (2022). Internal reflection ages, depths, and derived SMB are published in Cavitte et al. (2022). To summarize the method described in Kausch et al. (2020) and Cavitte et al. (2022) briefly, radar reflections are first dated using the ice core chronologies where they are closest to the ice core site. Radar reflection depths are measured in two-way-travel time from the surface. Conversion to real depth within the snowpack is done by iteration, by calculating the predicted reflection depths based on a best-fit Herron-Langway profile Herron and Langway (1980), the ages and depths of the dated reflections. The volume of snow accumulated between pairs of isochrones is then converted to an average SMB for the time period contained between each pair of isochrones. This is done for each radar data point of the radar surveys.

One drawback of this method is the necessity to hand pick reflectors from the GPR data. This means reflectors need to have a strong shift in permittivity, so that they are robust spatially and can be traced throughout the whole data set. Consequently, not all IRHs are suitable for tracking. Therefore the age of the first strongly reflective isochrone which can be traced through the majority of each ice rise is not always the same, resulting in different time periods for the SMB record of each ice rise (1993-2012 at DIR, 1982-2017 at LIR and 2008-2018 at HIR, Cavitte et al. (2022)).

### 3.3 Sentinel-1 angle correction

Since the cross-polarization ratio $\sigma_{HV}/\sigma_{HH}$ is heavily influenced by the satellite incidence angle (Fig. 4A showing R-value of -0.84 and a slope of -0.02 for all sample points), an angle correction algorithm was implemented to correct for the incidence angle variations, which vary between 20 and 40 degree for the available orbits. Therefore, we calculated the average cross-polarization ratio $\sigma_{HV}/\sigma_{HH}$ per SMB sampling location for the years 2014-2020 for every available orbit, where every

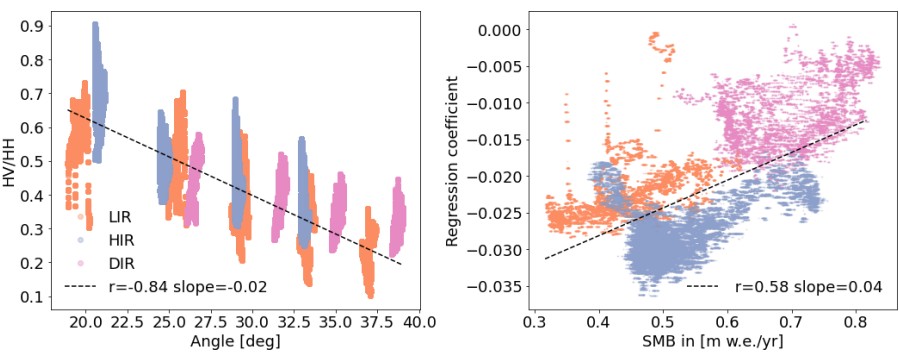

**Figure 4.** A) Angle dependency of the cross-polarization ratio. The average Sentinel-1 cross-polarization ratios for each sampling point from the three ice rises are color-coded, with blue being HIR, orange being LIR and pink being DIR. Each ice rise was covered with multiple orbits providing different incidence angles. B) shows the regression coefficient of the incidence angle and cross-polarization ratio correlation as a function of SMB. The points have the same color code as in A.

sampling point was covered by at least four different orbits. In this way, each sampling point was sampled under four different incidence angles, as each orbit represents a different incident angle. We calculated a linear regression for each sampling point based on the decrease of the cross-polarization ratio with incident angle. Using this linear regression we corrected the cross-polarization ratio for each pixel using:

$$\sigma_{\mathrm{HV}}/\sigma_{\mathrm{HH}}{}' = \sigma_{\mathrm{HV}}/\sigma_{\mathrm{HH}} - (-\beta_{\mathrm{point}} \times angle + \omega_{\mathrm{point}}) + (-\beta_{\mathrm{point}} \times 30 + \omega_{\mathrm{point}}) \tag{2}$$

Where $\sigma_{\mathrm{HV}} / \sigma_{\mathrm{HH}}{}'$ represents the angle-corrected cross-polarization ratio. $\beta_{\mathrm{point}}$ is the slope and $\omega_{\mathrm{point}}$ the intercept at one specific sampling point. Applying the angle correction individually to each pixel is necessary as the relationship between the incidence angle and the cross-polarization ratio varies spatially and also depends on the SMB (Fig. 4). Therefore, applying the angle correction per pixel removes any spatial variability noise in the incidence angle correlation. We normalized the values to a 30 degree incidence angle, as this is the central incidence angle of the available orbits. We did not correct for the surface slope or aspect of the topography as neither showed a clear relationship with the cross-polarization ratio. Furthermore, all ice rises have gentle surface slopes that do not exceed 1.4 °, which is small relative to the incidence angle variations of more than 10 °. In addition to the angle correction a spatial mean filter using a 3×3 pixel moving window was applied to reduce speckle noise.

## 3.4 Snowpack modelling

To understand which snowpack properties are responsible for Sentinel-1 backscatter intensity variability, we created several artificial snowpacks using the physical SNOWPACK model with meteorological forcing from AWS2. Since AWS2 data is only available for 2017-2019, it does not allow to construct a snowpack thick enough to correspond to the Sentinel-1 penetration

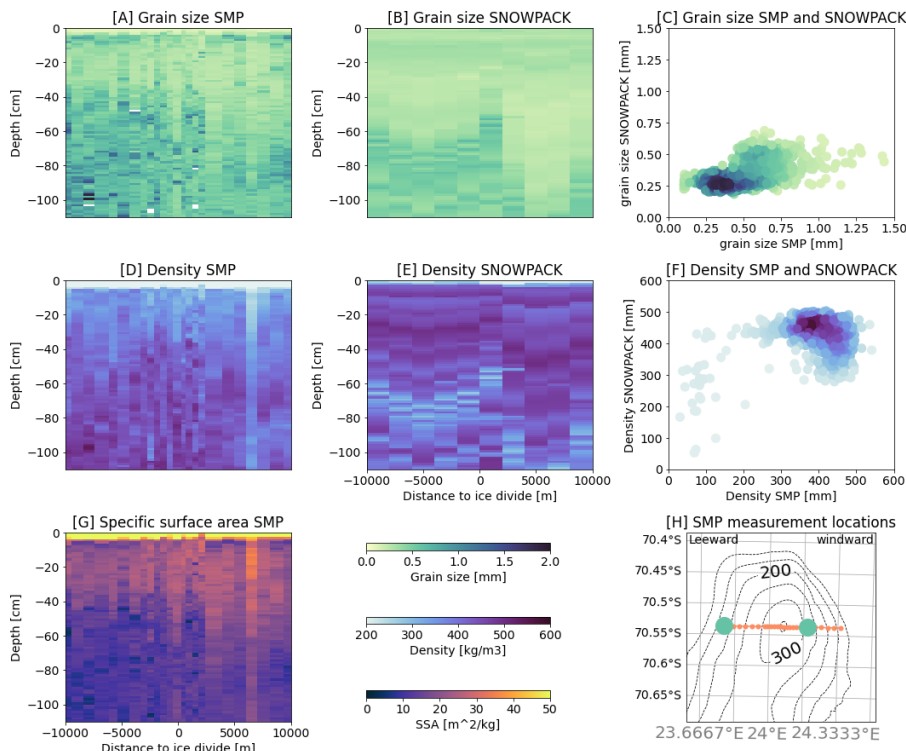

**Figure 5.** SMP measured optically equivalent grain diameter (A), density (D) and specific surface area (SSA) (G) at LIR as well as SNOW-PACK modelled optically equivalent grain diameter (B) and density (E). The third column shows how the results compare between SMP and SNOWPACK, with panel (C) showing optically equivalent grain diameter and (F) density. (H) shows the location of the SMP measurements taken across LIR (orange dots) as well as the locations of the AWS (green dots).

depth and corresponding volume scattering. For example, C-Band penetration depth on the Greenland ice sheet under comparable conditions was found to be around 12 -35 m (Zebker and Weber Hoen, 2000). Therefore, the SNOWPACK simulation was extended by re-using the 2018 input AWS2 for 20 years in a row at the beginning of the SNOWPACK simulation and extending it by the full 2018 to late 2019 AWS2 record, to ultimately create a 22 year long meteorological forcing file to run
5   SNOWPACK, where the first 20 years are the AWS2 measurements for the year 2018 on repeat and the last two years are the AWS2 measurements from 2018 to the end of 2019.

To create snowpacks that represent the spatial variability of snow along the SMB profile across the ice rise, we extended the AWS2 input data to 10 points across LIR (Fig. 5). This was achieved by using the same modelled snowpack from AWS2 but using the SMB reconstructed from the GPR data as input for precipitation. We used a GPR track across LIR, where we tracked
10  the 2002 layer (Fig. 2) and calculated the average yearly SMB in the same way as described in section 3.2. Furthermore, we assumed that this SMB profile is driven solely by snowfall, neglecting other accumulation processes such as the deposition of blowing snow.

To evaluate the modelled snowpacks, we used the SMP field measurements across the ice rise (Fig. 5). One of the key snowpack parameters for modelling backscatter is snow optically equivalent grain diameter. Figure 5 shows the modelled SNOWPACK values compared with the SMP measurements (Fig. 5 A, B, C). Figure 5 also shows that within the first 1 m of snow depth, SNOWPACK underestimates the optically equivalent grain diameter values observed by the SMP measurements near the surface. Furthermore Figure 5 highlights that SNOWPACK does not fully capture small-scale variability observed in the SMP data.

## 3.5   SMRT modelling

Finally we used the SMRT model to investigate the effect of increasing optically equivalent grain diameter on the cross-polarization ratio, as well as determining the effective penetration depth of the Sentinel-1 signal. We modelled the volume scattering of the snowpack in $\sigma_{HH}$ and $\sigma_{HV}$ for each Sentinel-1 observation date in time as well as for each additional modelled snowpack across the ice rise in space. The SMRT model (Picard et al., 2018a, b) is a microwave radiative transfer model for a multilayer snowpack for both active and passive sensors, which allows to model the radiation of different electromagnetic theories and microstructure models. We used the Improved Born Approximation (IBA) (Mätzler, 1998) together with the sticky hard sphere (SHS) microstructure representation (Picard et al., 2018b). However, a Dense Media Radiative Transfer (DMRT) model can also be used here instead of IBA as it is nearly equivalent for the SHS microstructure model (Löwe and Picard, 2015). In the IBA configuration together with SHS, the SMRT model requires density, layer thickness, optically equivalent grain diameter and stickiness as input data. Density, layer thickness and optically equivalent grain diameter are direct outputs from SNOWPACK. Stickiness can not be modelled by SNOWPACK and was set to 0.15 for all runs. Stickiness is a term that describes the degree of clustering of the snow grains. In SMRT a higher value for stickiness results in lower $\sigma_{HH}$ as well as $\sigma_{HV}$. However, $\sigma_{HV}$ decreases faster than $\sigma_{HV}$ with increasing stickiness resulting in a decrease of $\sigma_{HV}/\sigma_{HH}$ with stickiness. We chose 0.15 for stickiness as it represents a reasonable value for dry wind packed snow and it resulted in the best fit between SMRT and observed backscatter (SMRT API documentation recommends a value between 0.1 and 0.3 (Picard et al., 2023)).

To calculate the effect of increasing optically equivalent grain diameter on the signal, we used the modelled snowpack at the location of AWS2 and multiplied the optically equivalent grain diameter of each layer with an increasing factor ranging from 0.1 to 10. The resulting snowpack was then used as input for SMRT modelling (Fig. 9). To calculate the effective penetration depth, we started by running the SMRT model with only the most shallow layers of the snowpack and successively increased the number of layers used, by adding deeper and deeper layers with each run. We then defined that effective penetration depth is reached once the backscatter intensity does not further increase by adding deeper layers (Fig. 9).

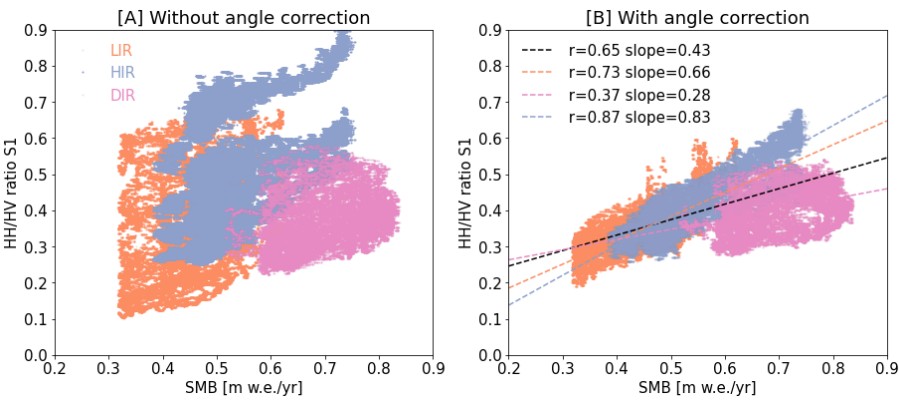

**Figure 6.** A) shows the empirical relationship between Sentinel-1 cross-polarization ratio and the SMB without incidence angle correction. At each point the SMB was reconstructed using the GPR data and the average cross-polarization ratio was sampled for that point. Orange points are located on LIR, blue on HIR and pink on DIR. B) shows the same but post incidence angle correction. The dotted black line shows the linear regression across all points, with an R-value of 0.65 and a regression coefficient of 0.43 with a standard error of 0.0006. The orange dotted line shows the linear regression using only points from LIR, with an R-value of 0.73 and a regression coefficient of 0.66 with a standard error of 0.0013. The blue dotted line shows the linear regression using only points from HIR, with an R-value of 0.85 and a regression coefficient of 0.87 with a standard error of 0.0009. The pink dotted line shows the linear regression using only points from DIR, with an R-value of 0.36 and a regression coefficient of 0.37 with a standard error of 0.0021.

## 4 Results

### 4.1 Empirical relationship between Sentinel-1 cross-polarization ratio and SMB

Figure 6A shows the relationship between the measured SMB and the uncorrected cross-polarization ratio of all available orbits for all three ice rises. The resulting plot shows little correlation between the SMB and the cross-polarization ratio, with an R-value of 0.20. After correction of the incidence angle, the correlation between the SMB and the cross-polarization ratio improves to an R-value of 0.63 (Figure 6 B). This improvement includes a modest increase in R-Value (~ 0.01) as a result of the speckle filter. However, it also becomes clear from figure 6 B that the relationship between the cross-polarization ratio and the SMB still differs between the ice rises, even after angle correction. LIR and HIR are fairly aligned with a variability of ~11% in regression coefficient after correction. However points from DIR differ more substantially with a variability of ~62% in regression coefficient, when compared to the average of LIR and HIR.

The cross-polarization ratio and SMB relationship can also be examined along GPR profiles across each ice rise (Fig. 7). Each profile shows a pattern of high cross-polarization ratio and SMB on the windward side and lower cross-polarization ratio and SMB on the leeward side. However, the three profiles also show that the cross-polarization ratio and SMB is most strongly correlated at HIR, where the two curves both show a north-south decline. At LIR, the profiles show generally the same pattern of high values on the eastern flank (windward side) and lower values on the western flank (leeward side) with a minimum at the

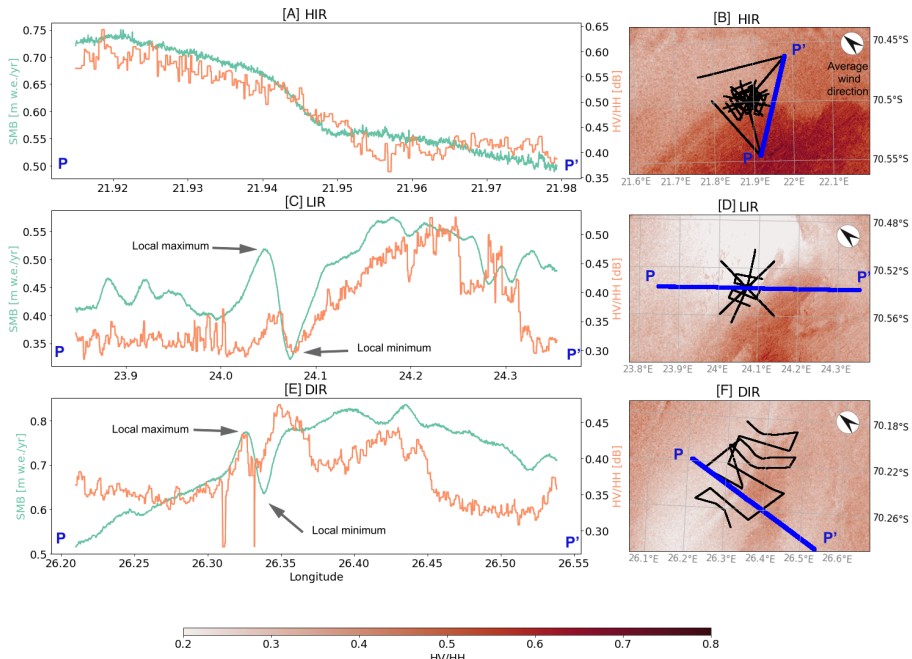

**Figure 7.** Panels A), C) and E) show a comparison between the cross-polarization ratio (in orange) and the SMB profiles (in green) at HIR, LIR and DIR, respectively. Each profile follows a GPR straight track. The location of the GPR tracks is shown in black panels B), D) and F). Panels B), D) and F) also show the spatial distribution of the cross-polarization ratio around the GPR tracks. The wind direction shown in B), D) and F) was measured by AWS2 and averaged between Dec 2017 and Dec 2019.

ice divide. However, at LIR we also observe a difference between the cross-polarization ratio and the SMB, for example at the eastern end of the profile where the cross-polarization ratio falls off while the SMB remains high. Similar differences occur at DIR where the cross-polarization ratio is generally lower on the windward side than the SMB, but increases at the western end of the profile, where SMB decreases. On the other hand, the cross-polarization ratio at LIR and DIR also matches smaller SMB

5  features, including the erosion driven SMB minimum at the peak of the ice rise and the deposition-driven SMB maximum just downwind of the peak (Fig. 7). Overall, the temporally-averaged cross-polarization ratio from all available orbits shows at least a partial correspondence to the SMB across all of the three ice rises, if an incidence angle correction is applied. However, there are local variations, where the cross-polarization ratio does not match the SMB, most notably on the windward flank of DIR.

## 4.2 Spatial and temporal variability of snow parameters

10  Using the SMP field measurements and the SNOWPACK model simulations, we have four parameters (density, optically equivalent grain diameter, SSA and the average layer thickness of the snowpack) to describe the snowpack micro-structure and analyse its spatial and temporal variability (Figs. 5, 8). Of these four, density and optically equivalent grain diameter are

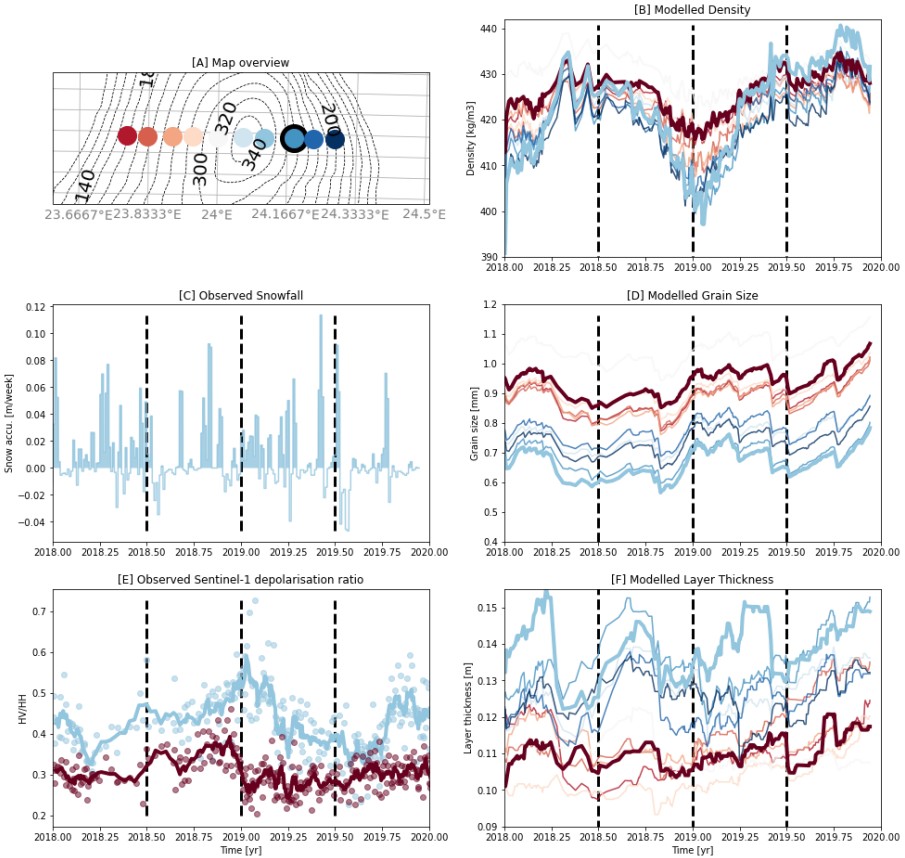

**Figure 8.** Observed and modelled time series across LIR. A) shows an overview map of the LIR. The black dot shows the location of AWS2, the colored dots represents the location of the time series shown in B) to F). B), D) and F) show density, optically equivalent grain diameter and layer thickness time series as modelled by SNOWPACK. Each line shows the average density, optically equivalent grain diameter and layer thickness of the first 15 m depth at each modelled time step. C) shows the snow accumulation per week measured by AWS2. E) shows the Sentinel-1 angle-corrected cross-polarization ratio on each side of the ice rise. A vertical dotted line marks every half year on panels B)-F).

available in both SMP data and SNOWPACK simulations, whereas in our dataset SSA is exclusive to the SMP data and layer thickness exclusive to SNOWPACK.

The spatial analysis shows that optically equivalent grain diameter and density are lower on the windward side, where accumulation is high, than on the leeward side where accumulation is low. This difference can be observed in both the SMP 5 data in the first meter of the snowpack (Fig. 5) and in the average density and optically equivalent grain diameter of the first 15 meter of snow depth in the SNOWPACK data (Fig. 8). Both of these differences in density and optically equivalent grain diameter are expected, as density and optically equivalent grain diameter increase with time, due to snow metamorphism (Leinss et al., 2020). So a constant influx of new snow which has generally low density and optically equivalent grain diameter,

decreases the average density and optically equivalent grain diameter within the first couple of meters. In addition, blowing snow redistributes snow from the windward flank of the ice rise to the leeward flank. Wind-redistributed snow typically exhibits a higher density than freshly fallen snow (Walter et al., 2024).

Similar spatial differences can be observed in the SSA data across LIR (Fig. 5). While SSA is not a direct measurement of grain shape, but it is influenced by it, as different grain morphologies result in different surface-to-volume ratios. Low SSA values generally correspond to larger, more rounded grains, whereas high SSA values are typically associated with smaller, more angular or faceted crystals. Fig. 10 shows higher SSA values on the windward side and lower values on the leeward side. The lower SSA values on the leeward side are likely a result of re−deposition of blowing snow. Blowing snow grains undergo wind-driven compaction, and as a result of that often end up as rounded grains when deposited (Vionnet et al., 2012; Walter et al., 2024).

The temporal analysis of the snowpack properties based on SNOWPACK model simulations shows that both optically equivalent grain diameter and density also show seasonality as a result of variations in temperature, wind speed and snowfall (Fig. 8, 2). Density is lower during the 2019 austral summer, when snowfall and temperatures are high and wind speeds are low. The difference between austral summer and winter density is stronger on the windward side of the ice rise than on the leeward side (Fig. 8). During austral winter, densities between windward and leeward side are comparable around 430 kg/m3. At the location of AWS2, density decreases to around 400 kg/m3 in austral summer, whereas density only decreases to around 420 kg/m3 on the leeward side. This can be explained by larger snowfall differences between windward and leeward side during austral summer months than during the winter. The SNOWPACK simulation shows moreover that optically equivalent grain diameters across the ice rise decrease in early 2018 and increase again at the end of the year. In 2019 the optically equivalent grain diameters remain high around 1.0 mm on the leeward side and increase further with the beginning of 2020. On the windward side optically equivalent grain diameters increase from 0.7 mm to 0.8 mm in 2019. The difference in modelled optically equivalent grain diameter between windward and leeward side does not change between winter and summer and remain constant in time.

The SNOWPACK time series also show that layer thickness is generally smaller on the leeward side, where SMB is low and larger on the windward side, where SMB is high. The layer thickness gradient can be explained by more frequent snowfall and less erosion on the windward side of the ice rise, creating thicker annual layers. Conversely more erosion and less frequent snowfall create smaller layers on the leeward side. The temporal variability of the layer thickness is higher on the windward side of LIR and lower on the leeward side. We do not observe seasonality in the layer thickness across LIR.

### 4.3 Optically equivalent grain diameter, density and their relationship to the cross-polarization ratio

Comparison of the field observations and SNOWPACK outputs with the spatial patterns of Sentinel-1 cross-polarization ratio shows that a high cross-polarization ratio more often occurs at locations where SMB is high (Fig. 7). Therefore, higher/lower cross-polarization ratios spatially coincides with lower/higher densities and lower/higher optically equivalent grain diameters, respectively (Fig. 10). This implies a negative correlation between cross-polarization ratio and density/optically equivalent grain diameter as this behaviour is consistent in space and can be observed in both the modelled SNOWPACK output (Fig.

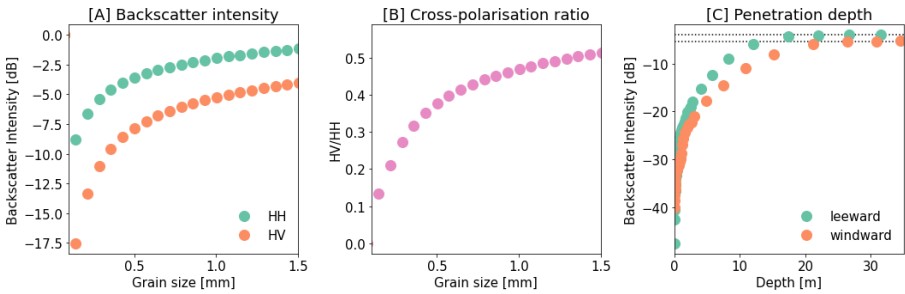

**Figure 9.** Synthetic backscatter intensity modelled with SMRT. A) shows the modelled $\sigma_{HH}$ and $\sigma_{HV}$ intensity with increasing optically equivalent grain diameter. B) shows the $\sigma_{HV}/\sigma_{HH}$ with increasing optically equivalent grain diameter. C) shows the increase in $\sigma_{HH}$ when including deeper layers on the windward and leeward side of LIR. The dotted lines in C) represent the Backscatter intensity of the complete snowpack, to highlight at which point the Backscatter intensity is no longer increasing.

8), as well as in the measured SMP values (Fig. 10). For example, the SMP data shows a negative-correlation between the density/optically equivalent grain diameter and the cross-polarization ratio with a Pearson correlation coefficient of 0.54/0.61 and a P-value of 0.0054/0.0013, respectively (Fig. 10).

Also the temporal analysis seems to suggest a negative correlation with density as the cross-polarization ratio differs most
between leeward side (around 0.6) and windward side (around 0.2) in the austral summer (differences up to 0.4; Fig. 8), when density variations between leeward and windward side are also maximal (up to 30 kg/m3). Whereas during austral winter both the polarization differences (around 0.1 - 0.2) and density differences (around 5 to 10 kg/m3) between windward and leeward side are generally also lower. This results in little seasonality of the cross-polarization ratio on the leeward side, where values hover around 0.3 throughout the whole year vs. strong seasonality on the windward side, where values range from ∼0.3 to 0.6.
This again, is mirrored in the density where there is little seasonal change on the leeward side, and strong seasonal change on the windward side.

Since both the field measurements and SNOWPACK output suggest a negative correlation between cross-polarization ratio and optically equivalent grain diameter, we tried to recreate this by running SMRT simulations with varying optically equivalent grain diameter. However, this negative correlation between optically equivalent grain diameter and cross-polarization ratio
cannot be reproduced by the SMRT radiative transfer model as the simulations show an increasing cross-polarization ratio with an increasing optically equivalent grain diameter (Fig. 9). Additionally, SMRT shows higher effective penetration depth on the windward side than on the leeward side. On the windward side 90 % of the signal originates from the first ~ 20 m, whereas on the leeward side 90 % of the signal originates from the first ~ 17 m.

## 5  Discussion

Within our study area of Dronning Maud Land, we found a spatial correlation between Sentinel-1 cross-polarization ratio and the SMB obtained from field GPR measurements (Fig. 6, 7). Areas of high SMB show a high cross-polarization ratio and

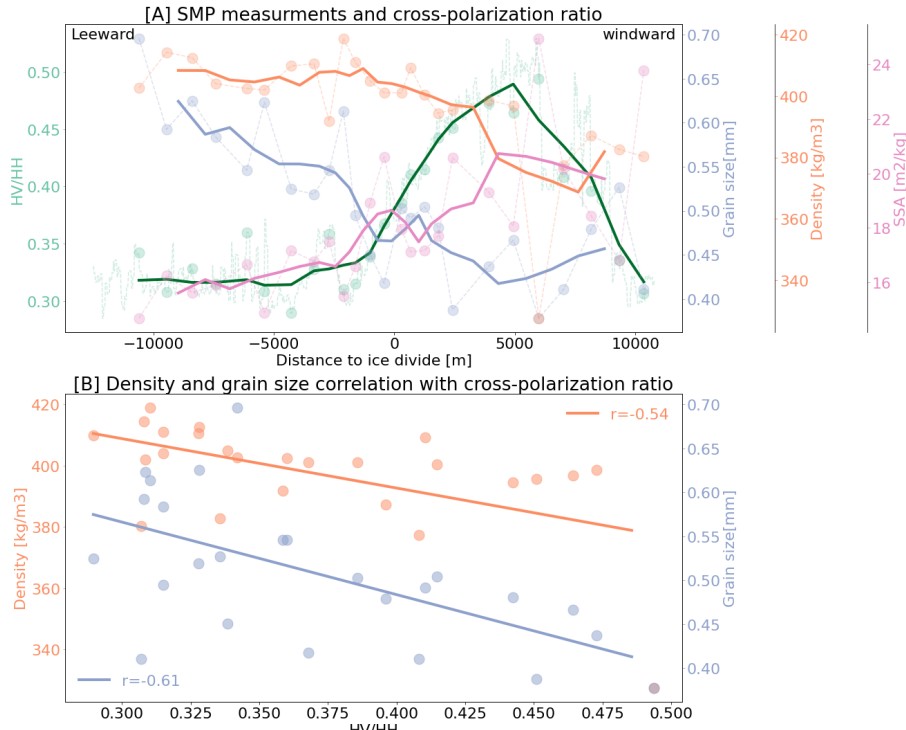

**Figure 10.** A) SMP measurements compared to the cross-polarization ratio across LIR in a east to west profile across the LIR. SMP measurements include the average optically equivalent grain diameter, density and SSA of the first meter of snow. The points are the actual measurements and the solid lines are running means with a 4 point window for the SMP measurements and a 80 pixel running mean for the Sentinel-1 cross-polarization ratio. The location of the SMP measurements is shown in Fig. 5 H. B) shows the correlation between the the cross-polarization ratio and the optically equivalent grain diameter (blue) and the density (orange) measured by the SMP.

vice versa. Additionally, our SNOWPACK modelling and SMP field measurements show that these areas of high SMB are accompanied by low optically equivalent grain diameter and density as well as high SSA (Fig. 10). optically equivalent grain diameter and density are low most likely because of the larger amount of snowfall on the windward side, whereas on the leeward side, densities and optically equivalent grain diameters are higher and the SSA of individual grains is lower. This can

5  be explained as a result of snowpack densification. In high SMB regions, frequent deposition of fresh snow maintains lower density snow with smaller optically equivalent grain diameters. On the other hand, on the windward side, low SSA indicates a rounded grain shape as a result of wind-driven compaction, which also explains higher densities and optically equivalent grain diameters (Keenan et al. (2022), Wever et al. (2023)).

    The SMRT model results show that higher optically equivalent grain diameters and densities on the leeward side of LIR can

10  explain increases in both co-polarization and cross-polarization (Fig. 9). However, the model also predicts an increase in cross-polarization ratio with larger optically equivalent grain diameters, which is inconsistent with the Sentinel-1 measurements showing lower cross-polarization ratios on the leeward side of LIR where optically equivalent grain diameters are high (Fig.

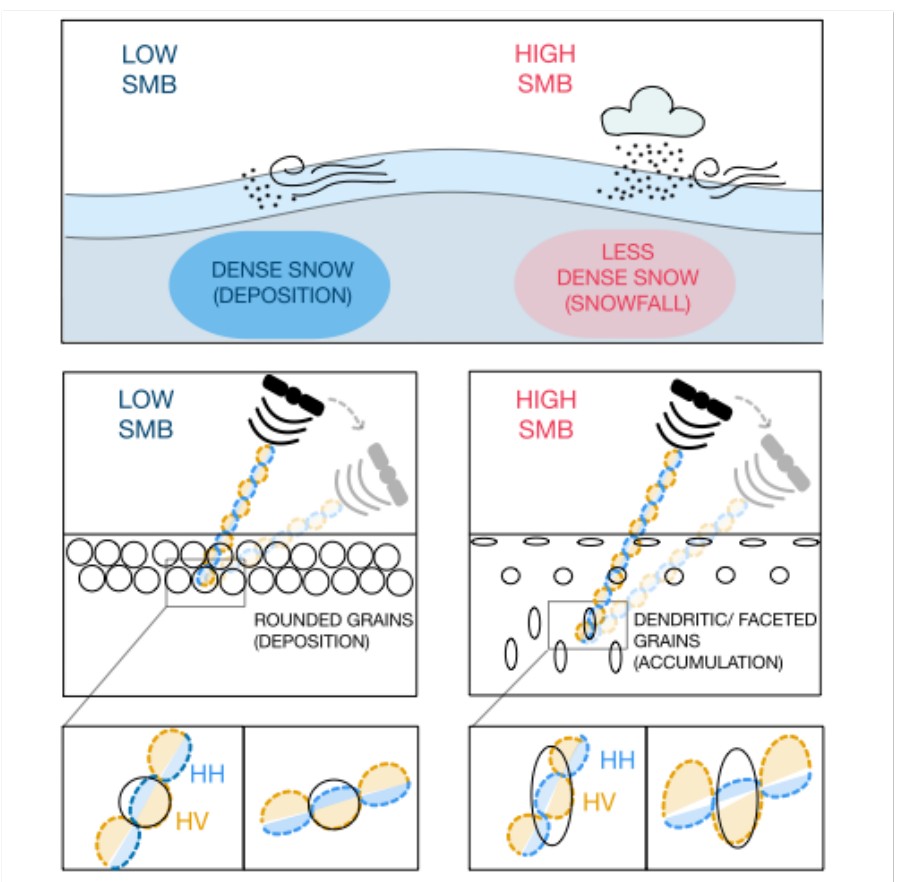

**Figure 11.** Sketch illustrating the difference in snow micro structure and its volume scattering response for the windward and leeward sides of an ice rise. The upper panel describe the macroscopic difference in accumulation, whereas the lower panels illustrate the microscopic differences in grain shape and their effect on backscatter incidence angles.

10). The SMP data further indicate a negative correlation between density or optically equivalent grain diameter and cross-polarization ratio. Therefore, while large optically equivalent grain diameters on the leeward side of the ice rise coincide with low cross-polarization ratios, they are unlikely to be the primary cause. This interpretation is, however, limited by the fact that SMP measurements extend only to 1 m depth, whereas scattering contributing to the cross-polarization ratio may occur below

5   this depth.

A variety of explanations can be given for the difference between observation and modelling of cross-polarization differences. First, the difference could be explained by SMRT not properly accounting for anisotropy and assuming spherical grains. However, the SMP field measurements show that the snow grains on the windward side of the LIR are not spherical, as indicated by their high SSA (Fig. 10). Fresh snow is known to be anisotropic and generally starts horizontally aligned and becomes

10   more vertically aligned with time due to temperature gradient metamorphism (Leinss et al., 2020). Consequently, the vertical

alignment of grains in high SMB areas could be a second explanation but it contradicts with the lower angle dependency for the cross-polarization ratio in high SMB areas (Fig. 4). Stronger vertical alignment of grains should result in stronger dependency on incidence angle, which counteracts the decrease of cross-polarization ratio with angle due to less surface scattering (Fig. 11), at least when not considering multiple-scattering. A vertically aligned grain has its largest extent in the direction of the wave propagation when looking from a near nadir angle. However, the radar wave is only sensitive to grain extent perpendicular to its travel path. Therefore the vertically polarized part of the radar wave only becomes sensitive to the large vertical extent if the angle increases away from nadir (Fig. 11).

Thirdly, surface scattering could be an explanation for the relation between SMB and cross-polarization ratio. Intuitively one might assume that high snowfall in an area of high SMB would decrease surface roughness, whereas wind erosion would decrease SMB in an area of low SMB. However, a recent study by Studinger et al. (2020) did not find a statistically significant relationship between surface roughness and accumulation rates. While this cannot rule out that surface scattering plays a role, it renders it unlikely that the observed empirical relationship between SMB and cross-polarization ratio is driven by surface scattering.

Lievens et al. (2019) explain their empirical relationship between snow depth in the northern hemisphere and the cross-polarization ratio with the longer travel path of the radar signal through the snowpack with increasing total snow column height. An analog for this on the Antarctic ice sheet could be the effective penetration depth. Lower densities and lower optically equivalent grain diameters in an accumulation area result in a higher effective penetration depth (Fig. 9). This results in a longer travel path of the radar signal through the snowpack, increasing multiple scattering as well as anisotropic scattering, akin to Lievens et al. (2019) explanation.

Therefore a fourth possible explanation for the correlation between cross-polarization ratio and SMB, could be a combination of effective penetration depth, grain shape, erosion and deposition and surface scattering. First, high SMB on the windward side allows for high effective penetration depths. Second, due to accumulation being mostly by snowfall, the snow is anisotropic, at least when considering freshly fallen snow. Third, the radar signal reaches the vertically aligned snow grains due to the higher effective penetration depth, and thus a higher cross-polarization ratio is achieved. On the other hand, on the leeward side, low effective penetration depth and rounded grains, which are due to more deposition, create the opposite effect (Fig. 11).

What remains unclear is whether or not this relationship could be applied outside of the study area presented here. All GPR tracks used in this study to reconstruct the SMB were recorded in a similar environment within a couple of hundred kilometres from each other. All three ice rises represent dry snow accumulation zones, which are a result of orographic uplift from steady katabatic winds. Areas with surface melting, or with a very low accumulation rate like the Antarctic plateau, might not behave in the same way. Therefore, further testing of this relationship with spatially distributed SMB data sets that cover diverse locations across Antarctica will be necessary to test this. However, reliable, recent, spatially distributed SMB measurements from Antarctica are still far and few in comparison. The advantage of this study area is that there is a unique data set available of multiple GPR tracks all accompanied by ice core dating and none of them older than 2012.

Here we argue that our study can be seen as a proof of concept, that shows that Sentinel-1 cross-polarization can be used as a proxy for surface mass balance under the right circumstances. This provides a great opportunity for Antarctic mass balance

research in the future, as the satellite SAR data library will only grow and become more extensive with time. Additionally, any field data collected now always has coeval satellite image with it. This will ease the process of evaluating the usage of cross-polarization ratio as an SMB proxy Antarctica-wide.

# 6 Conclusions

Here we used SMB field measurements across three ice rises in Dronning Maud Land to establish an empirical relationship between the SMB and the Sentinel-1 cross-polarization ratio with an R-value of 0.63. This empirical relationship between the SMB and the cross-polarization ratio is only observed after correcting for incidence angle. Additional SMP field observations and modelling results suggest that a combination of effective penetration depth variability and grain shape variability is driving this relationship. However, more field data is needed to strengthen this argument. Furthermore, we observe a spatial pattern across all three ice rises. The windward high snow accumulation flank of the ice rise has a higher cross-polarization ratio compared to the leeward flank. These spatial differences are more pronounced during austral summer and weakened during austral winter. Overall, the findings are promising and highlight the potential of SAR remote sensing for an initial assessment of SMB. The ability to measure recent SMB from satellite measurements would greatly facilitate and speed-up the process to evaluate mass balance at the continental scale.

*Author contributions.* MGPC provided the GPR SMB datasets. TK and EK provided the SMP dataset. SL set up the AWS and TK and EK recovered them. TK prepared the paper with contributions from all co-authors.

*Competing interests.* Stef Lhermitte is a member of the editorial board of The Cryosphere.

*Acknowledgements.* Thore Kausch and Stef Lhermitte were supported by the NWO Polar Program (ALWPT.2016.4). Marie Cavitte is a postdoctoral researcher of the FRS-FNRS. We would like to thank Ghislain Picard, Melody Sandells and Henning Löwe for developing and providing SMRT. Finally we thank the International Polar Foundation (IPF) for the logistic support during the Mass2Ant field campaigns.

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
