# Peer review of "Sentinel-1 cross-polarization ratio as a proxy for surface mass balance across east Antarctic ice rises"

_EGUsphere, 2024_

## Referee Comment (RC1)

**Review of Manuscript 2024-EGUSphere-2024-2077:**
**Sentinel-1 cross-polarization ratio as a proxy for surface mass balance across east Antarctic ice rises**

William David Harcourt, University of Aberdeen

September 25, 2024

**Overview**

This study develops an empirical relationship between Surface Mass Balance (SMB) and the cross-polarisation ratio ($\sigma_{HV}/\sigma_{HV}$) of Sentinel-1 SAR data derived from a comparison to in situ snow data acquired from Antarctica. The relationship has been shown to demonstrate clear accumulation patterns across the three ice rises studied. On the windward side, high SMB and cross-polarisation ratio is associated with low density snow and smaller grains, whilst the opposite is true on the leeward side. A key part of the data processing is correcting for the satellite incidence angle which the study computes using a linear regression between SMB and at sampled locations and cross-polarisation ratio. Mapping these patterns shows that this ratio may be used as a proxy for SMB across Antarctica and therefore has potential to help map SMB in areas with few in situ measurements. Because of the potential application of this method, I believe the study should be published subject to the revisions below.

**General Comments**

My general comments can be split into three points:

- I am unsure about how the incidence angle correction has been implemented. Sentinel-1 has an incidence angle file associated with it, which I assume you use (although this is not stated). What is the result if you apply a standard approach such as conversion to $\gamma^0$ or $\sigma^0$ (divide by $\sin\theta$) (e.g. Small 2011). As I understand it, the regression coefficients used in Eq. 2 are derived from a regression between the cross-polarization ratio and angle. But you say this is calculated at each sampling point, so it's not clear what data is being used. Rewriting some of the text in Section 3.3 will probably help to clarify these points.

- Much of the results section discusses results as if there is a clear pattern between e.g. SMB and cross-polarisation ration. Whilst I can see there is a relationship, to me the pattern is variable and not consistent, implying there is more complex physics at play. Downplaying some of the results and emphasising the variable due to e.g. snowfall variations, local climate might help with this.

- The text is a bit colloquial in places. Phrases such as 'want to' and 'coming from' and 'steady' are used which do not describe some of the underlying processes being discusses e.g. quantifying wind speeds, describing the travel orientation of winds. Editing the text throughout will help

here.

**Technical Corrections (References to page numbers (P) line (L) numbers in preprint)**

**Abstract**

P1L3: 'large spatial coverage and and ability to penetrate the snowpack'

P1L15: You probably want to add that the proxy should be combined with physical models.

**Introduction**

P1L19: 'large uncertainties': how large are the uncertainties? Maybe quote mass balance for year e.g. 2023 + uncertainty?

P2L6-7: This sentence repeats what you've just said 'as in-situ measurements are sparse'. Suggest remove?

P2L13: Also different densities of dry snow, wet snow, firn and ice.

P2L26: Is this because the increase in travel time due to snow thickness increases is larger for the co-polarised image (i.e. $\sigma_{VH}$ becomes larger than $\sigma_{VV}$? I think this should be stated clearly.

P2l29: 'ground that ice not ice'?

P2L33: Remove 'want'

P2L35: 'driving the cross-polarization ratio variability, which relates to volume scattering from the snowpack.'

P3L7: Change 'synthetic' to 'theoretical'

**Data and study area**

P3L17: Not keen on 'island-like topography'; maybe 'a protruding bedrock bump' or something similar?

P3L25: ', this allows for'

P3L26: 'crevassing which creates a strong surface scattering response, the backscatter signal from the undisturbed snowpack will dominate.'

P3L26: Can you comment on the accuracy of ERA5 for interpolating? How well does it match AWS1 data? Also state pixel size.

P4L15: I would include the tracks on Fig.1 if you can.

P4L18-19: State that the dating using ice cores is described below.

P4L19: Same GPR system as before? If not, please describe it briefly.

P4L26: Can you briefly explain of SSA was calculated given that it is included in Eq. 1 below?

P6L1: I would include the locations of these samples in Fig. 1.

P6L4-6: Remove repitition (e.g. Sentinel-1). How well distributed were the images across the year? E.e did you have images in particular seasons? Could you also state why you average across 6 years -

my assumption is to remove noise, but snow conditions might change quite significantly from year to year.

P6L9: Just to be absolutely clear, I read 'the logarithmic ratio between' as the ratio between $\sigma_{HV}$ and $\sigma_{VV}$ in dB?

**Data and study area**

P6L24: 'to what degree'

P7L3: Is the space between isochrones only related to snow accumulation? What about firn, ice or even ice lenses (maybe not an issue here given the low melt rates)?

P8L14-19: Combine with paragraph above.

P8L11: I would suggest clearly stating that the linear regression is between cross-polarization ratio and incidence angle (taken from the Sentinel-1 data set).

P8L16: Not sure this makes sense to me 'obscuring the incidence angle correction'

P9L6-8: Not sure I understand how the AWS2 data was extended by 20 years? What does it mean by 'looping'?

P9L9-P10L2: This section isn't clear to me e.g. how the AWS data was extended.

P10L6: Which panel are you referring to? Also, visually Fig, 4d and e don't align well together, yet you state in the text they do?

**Results**

P10L25: 'measured SMB' - important to emphasis what is measured and what is modelled.

P11L1-12: The patterns described in this section are a little vague and I struggle to follow some of it. For example, you state that SMB and cross-polarisation is higher on the windward side, lower on the windward side, but visually this does not match the graphs, they are very variable. Fig. 6A is most clear, so I think you can make the case for this pattern here, but for C and E I would instead on the variability of the pattern. It's still okay to state the broad pattern, but I would refrain from saying it is 1clear'.

P11L2: Could you label where the windward side is on the profiles should be for clarity?

P13L4: Again, best to annotate windward and leeward side throughout on you figures.

P14L7-13: Similar to above, the patterns are not totally clear. I would suggest being more cautious in your description.

**Discussion**

P16L1-2: This doesn't explain the reason why grain sizes are high on the leeward side & low on the windward side? I would think high SMB would mean higher density due to greater snow compaction? I am possibly misinterpreting things.

P17L33: 'Antarctic'

P18L3-4: What are those 'right circumstances'?

**Figures**

Figure 2: I find it a little confusing to have a distance profile on top then time series below, can you make 2 separate figures?

Figure 3: What do the dots represent? Average HV/HH for each pixel?

**References**

Small, D. (2011), 'Flattening gamma: Radiometric terrain correction for sar imagery', *IEEE Transactions on Geoscience and Remote Sensing* **49**(8), 3081–3093.

---

## Editor Decision (ED1)

**Review of Manuscript 2024-EGUSphere-2024-2077: Sentinel-1 cross-polarization ratio as a proxy for surface mass balance across east Antarctic ice rises**

William David Harcourt, University of Aberdeen September 25, 2024

**Overview**

This study develops an empirical relationship between Surface Mass Balance (SMB) and the cross- polarisation ratio ($\sigma_{HV}/\sigma_{HV}$) of Sentinel-1 SAR data derived from a comparison to in situ snow data acquired from Antarctica. The relationship has been shown to demonstrate clear accumulation patterns across the three ice rises studied. On the windward side, high SMB and cross-polarisation ratio is associated with low density snow and smaller grains, whilst the opposite is true on the leeward side. A key part of the data processing is correcting for the satellite incidence angle which the study computes using a linear regression between SMB and at sampled locations and cross-polarisation ratio. Mapping these patterns shows that this ratio may be used as a proxy for SMB across Antarctica and therefore has potential to help map SMB in areas with few in situ measurements. Because of the potential application of this method, I believe the study should be published subject to the revisions below.

**General Comments**

My general comments can be split into three points:

- I am unsure about how the incidence angle correction has been implemented. Sentinel-1 has an incidence angle file associated with it, which I assume you use (although this is not stated). What is the result if you apply a standard approach such as conversion to $\gamma^0$ or $\sigma^0$ (divide by sin θ) (e.g. Small 2011). As I understand it, the regression coefficients used in Eq. 2 are derived from a regression between the cross-polarization ratio and angle. But you say this is calculated at each sampling point, so it's not clear what data is being used. Rewriting some of the text in Section 3.3 will probably help to clarify these points.

- Each sample point was covered by at least 4 orbits. The incidents angles for each orbit vary (somewhere between 20 and 40 degree (Fig. 3A)). So the

- different orbits are whats used to derive the regression. We will rewrite Section 3.3 and try to clarify this.

- P9 L1-2

- Much of the results section discusses results as if there is a clear pattern between e.g. SMB and cross-polarisation ration. Whilst I can see there is a relationship, to me the pattern is variable and not consistent, implying there is more complex physics at play. Downplaying some of the results and emphasising the variable due to e.g. snowfall variations, local climate might help with this.

- We agree that there are more complex physics at play and the paper is merely trying to look at correlations and discuss possible mechanisms that could drive them. We don't think that we have a solid physical explanation for the observations and we will try to make it more clear that we are not claiming to have one.

- P12 L11, P13 L8

-

- The text is a bit colloquial in places. Phrases such as 'want to' and 'coming from' and 'steady' are used which do not describe some of the underlying processes being discusses e.g. quantifying wind speeds, describing the travel orientation of winds. Editing the text throughout will help here.

- We will rewrite the text to ensure it has a more professional tone, replacing colloquial phrases with more formal language to accurately describe processes.

**Technical Corrections (References to page numbers (P) line (L) num- bers in preprint)**

We will accept all technical corrections and reply to the questions in the following notes.

Abstract

P1L3: 'large spatial coverage and and ability to penetrate the snowpack'

P1 L3

P1L15: You probably want to add that the proxy should be combined with physical models.

Introduction

P1L19: 'large uncertainties': how large are the uncertainties? Maybe quote mass balance for year e.g. 2023 + uncertainty?

Will do so.

P2L6-7: This sentence repeats what you've just said 'as in-situ measurements are sparse'. Suggest remove?

The second sentence is meant specifically towards AWS's, snow stakes and firn cores, whereas the first one is meant generally towards all in-situ measurements. So I don't think its completely redundant.

P2L13: Also different densities of dry snow, wet snow, firn and ice.

P2L26: Is this because the increase in travel time due to snow thickness increases is larger for the

co-polarised image (i.e. $\sigma_{VH}$ becomes larger than $\sigma_{VV}$? I think this should be stated clearly.

We will rewrite the sentence to make it more clear.

P2l29: 'ground that ice not ice'?

This comment is unlcear to me,

P2L33: Remove 'want'

P2 L34

P2L35: 'driving the cross-polarization ratio variability, which relates to volume scattering from the snowpack.'

P3 L4

P3L7: Change 'synthetic' to 'theoretical'

P3 L11

Data and study area

P3L17: Not keen on 'island-like topography'; maybe 'a protruding bedrock bump' or something similar?

Personally I would keep 'island-like topography' as a protruding bedrock bump' seems overly complicated.

P3L25: ', this allows for'

P3 L29

P3L26: 'crevassing which creates a strong surface scattering response, the backscatter signal from the undisturbed snowpack will dominate.'

P3 L32

P3L26: Can you comment on the accuracy of ERA5 for interpolating? How well does it match AWS1 data? Also state pixel size.

We will add a sentence for that.

P4 L18-19

P4L15: I would include the tracks on Fig.1 if you can.

P4L18-19: State that the dating using ice cores is described below.

P4L19: Same GPR system as before? If not, please describe it briefly.

Yes, same system. We will add a sentence noting that.

P4 L26

P4L26: Can you briefly explain of SSA was calculated given that it is included in Eq. 1 below?

We got the SSA form the SMP measurement using the empirical approach explained in Proksch et al. (2015). Is this what you mean, or do you mean what SSA is in general? In this case We can add a short sentence explaining that.

P5 L2, P6 L1-2

P6L1: I would include the locations of these samples in Fig. 1.

P6L4-6: Remove repitition (e.g. Sentinel-1). How well distributed were the images across the year? E.e did you have images in particular seasons? Could you also state why you average across 6 years -

my assumption is to remove noise, but snow conditions might change quite significantly from year to year.

We will add an overview of the seasonal distribution of the sentinel-1 data.

P6L9: Just to be absolutely clear, I read 'the logarithmic ratio between' as the ratio between $\sigma_{HV}$ and $\sigma_{V\,V}$ in dB?

Yes.

Data and study area

P6L24: 'to what degree'

P7 L5

P7L3: Is the space between isochrones only related to snow accumulation? What about firn, ice or even ice lenses (maybe not an issue here given the low melt rates)?

This approach is based on the assumption that the snowpack follows the the Herron–Langway firn densification model. So that we can model the density of each layer and use that for the accumulated mass. Therefore ice lenses would indeed be problematic, but as you are saying that is not an issue here due to the low melt rates.

P8L14-19: Combine with paragraph above.

P8L11: I would suggest clearly stating that the linear regression is between cross-polarization ratio and incidence angle (taken from the Sentinel-1 data set).

P8L16: Not sure this makes sense to me 'obscuring the incidence angle correction'

P9L6-8: Not sure I understand how the AWS2 data was extended by 20 years? What does it mean by 'looping'?

The record of the same year (2018) was used repeatedly.

P9L9-P10L2: This section isn't clear to me e.g. how the AWS data was extended.

P10L6: Which panel are you referring to? Also, visually Fig, 4d and e don't align well together, yet you state in the text they do?

Line 6 refers to panel 4C. In the text we wanted to argue that 4a and 4b align well not 4d and e. Since we are talking about grain size there. However upon re-reading the paragraph we agree that this is not clear at all from the text and understand the confusion. We will rewrite the paragraph accordingly.

P9 L31

Results

P10L25: 'measured SMB' - important to emphasis what is measured and what is modelled.

P12 L3

P11L1-12: The patterns described in this section are a little vague and I struggle to follow some of it. For example, you state that SMB and cross-polarisation is higher on the windward side, lower on the windward side, but visually this does not match the graphs, they are very variable. Fig. 6A is most clear, so I think you can make the case for this pattern here, but for C and E I would instead on the variability of the pattern. It's still okay to state the broad pattern, but I would refrain from saying it is 1clear'.

We will remove the word "clear".

P11L2: Could you label where the windward side is on the profiles should be for clarity? P13L4: Again, best to annotate windward and leeward side throughout on you figures.

We will do so.

P14L7-13: Similar to above, the patterns are not totally clear. I would suggest being more cautious in your description.

P14 L32

**Discussion**

P16L1-2: This doesn't explain the reason why grain sizes are high on the leeward side & low on the windward side? I would think high SMB would mean higher density due to greater snow compaction? I am possibly misinterpreting things.

Fresh snow has low density and grain size. With time snow compaction will increase the density and grain size. This means that snow near to the surface will have lower density and grain size in areas of high SMB where the snow had less time for compaction.

P17L33: 'Antarctic'

P18 L23

P18L3-4: What are those 'right circumstances'?

See P17 L32-34: "All three ice rises represent dry snow accumulation zones, which are a result of orographic uplift from steady katabatic winds. Areas with surface melting, or with a very low accumulation rate like the antarctic plateau, might not behave in the same way."

Figures

Figure 2: I find it a little confusing to have a distance profile on top then time series below, can you make 2 separate figures?

We will try to separate them more clearly.

Figure 3: What do the dots represent? Average HV/HH for each pixel?

The linear regression between the variables.

**References**

Small, D. (2011), 'Flattening gamma: Radiometric terrain correction for sar imagery', IEEE Trans- actions on Geoscience and Remote Sensing 49(8), 3081–3093.

P2. Line 16, The authors claim that SAR is sensitive to snow microstructure and is independent of cloud cover. In fact, studies show that cloud cover can impact the radar response and that the SAR the sensitivity to snow microstructure is frequency-dependent. I would encourage the authors to be more precise in their writing.

Thank you for mentioning this. We will include this information.

P1 L19

P2 line 33. Using the co-pol HH channel will have less sensitivity to volume scattering-dominant processes that are evident in the VV co-pol channel, as used by Lievens et al (2018). Likely this may impact the results so the authors should explain more what the impact might be.

We will add some further explanation on that.

P2 L35
P3 L1-2

P3 Line 15. The locations of the LIR, HIR and DIR are not labelled in Fig 1 making this quite difficult to assess and understand what leeward/windward actually means.

Yes I agree, the ice rises should be labeled in Fig. 1. We will add labels.

P5 Fig. 1

P4 Line 11. Since ERA5 was used to gap fill in 2018, the authors should provide an assessment of the uncertainty of this gap-filling since wind speed has a significant impact on the SNOWPACK estimates.

We are confident in the ERA5 wind data as it is in good agreement with the AWS wind speeds in 2019 (Figure 2). However we are open to the idea of including a quantitive uncertainty assessment of the ERA5 data.

P4 L18-19

P4 eq 1. Please define all units used.

We will do so.

P5 L4 Eq. 1

P4 Line 27, Equation (1) and P5 lines 1-5 and throughout the paper. The authors should be more specific in their language referring to "grain size". This variable is a critical parameter in the SMRT estimation process and there are several emerging terms regarding what is meant by grain size (effective grain size, measured or observed grain size, optical grain size). Also, the concept of a "real grain size" is somewhat misleading.

We agree that "grain size" is an imprecise name. However further along in the text it is explained what exactly is meant by that (P5,L2-3). So I think the best nomenclature I could think of would be "estimated grain size"?

EXTRA NOTE:
Changed to optically equivalent grain diameter throughout the manuscript

P5 Fig 1. Does the scale bar for B (HH pol) apply to the cross-pol in C? This should be stated. Also, the GPR tracks are not clear - the authors provide a more detailed map of these tracks at the 3 locations.

Yes the scale bar does apply to Fig. 1C as well, we will make that more clear.

The tracks shown in the figure are the same tracks as shown later. The problem is that this is a more zoomed out view making them hard to see. I think form a graphic design point of view it will unfortunately be difficult to have them clearly visible on this zoom level without dramatically increasing the size of the figure. However we will try to improve this as much as possible.

P5 Fig. 1 Caption

P6 lines 14-15. 50 m spatial resolution is much finer resolution than the Lievens (2019) approach. They have noted in their paper that the smoothed backscatter data was posted to 1 km. Why did you select 50 m?

We chose a higher spatial resolution for the Sentinel 1 data since we also have higher resolution data available from the GPR for the correlation. This allowed us o maximise the number of data points available for analysis.

P6 Section 2.5. Did the authors include speckle filtering in their workflow? Even for EW data, speckle noise may have an impact and when the data are averaged, the speckle (multiplication noise) could have an impact on the averaging process of the S1 data. How do they know that this doe not have an effect?

No speckle filtering was included. We will analyse the impact of a speckle filter on the results.

P9 Line 7. Can the authors explain what they mean by "looping the 2018 input AWS for 20 years"?

The record of the same year (2018) was used repeatedly.

P10 Line 3-6. I disagree that there is "good agreement between SNOWPACK grain radius (?) and the SMP snow grain radius (?). There is much more variability in the SMP data than observed by the model indicating a lack of model sensitivity. Can the authors explain what this might be caused by and the importance of this?

We agree that the term "good agreement" is subjective and we will remove it. It is however expected that SNOWPACK does not fully capture small-scale variability observed in the SMP data. We will clarify this in the manuscript by noting that such variability is not represented in the model.

P9 L31-31

P10 Line 21. Penetration depth in microwave research is defined as 1/e. Is this what the authors mean or do they mean the maximum depth beyond which no further response is observed?

The maximum depth beyond which no further response is observed.

EXTRA NOTE:
Changed penetration depth to effective penetration depth throughout the manuscript

P10 Section 4.1. I know this is pedantic but the authors seem to conflate Correlation R with coefficient of determination ($R^2$) which is the measure of the fit of a linear regression. Perhaps they can be consistent in their use of such standard terms.

Figure 5 should include a legend of the colours for improved clarity.

We will add a colour legend.

P11 Fig. 5

P10 Section 3.5. The authors state that they use a stickiness value of 0.15 for all runs. How was this value selected and how sensitive are the results to it?

Generally speaking both hh and hv decrease with increasing stickiness, with hv decreasing faster, resulting in a generally lower hv/hh ratio with increasing stickiness. However, since the same stickiness is used everywehere, it does not

introduce any relative variability along track and therefore has little impact on the correlation. Nether the less we will add some more background on this to the method part.

P11 L1-4

P10 Section 3.5. Why did the authors select the IBA and not, for example the DMRT approach. It would be helpful for the reader to provide this justification. Furthermore, what was the substrate condition used in the model - was it an infinite background somehow? A more comprehensive explanation of the model set-up would certainly help the reader follow the logic here.

We will asses the impact different settings of the radiative transfer model on the results.

P10 The authors should include standard error metrics of the regression lines (the slope coefficient). What is the variability of the regression coefficients calculated? And how is this calculated?

We will include  standard error metrics.

P11 Fig. 5 Caption

P11/12. The role of Figure 6 is unclear. I understand it shows the SMB variations with cross-pol ratio but the patterns cannot be explained easily, despite the authors asserting that correspondence between SMD and cross-pol ratio is "clear". I can see that there is correspondence between the SMB and the cross-pol ratio for the HIR but for the LIR it is somewhat related but the DIR has only a moderate correspondence. It is unfortunate that in situ data are not available for the DIR and especially the HIR location where there is indeed the strongest agreement. The authors conduct an analysis of LIR based on the SMP, SNOWPACK, SMRT and cross-pol data. But no similar analysis can be undertaken of DIR and HIR because no microstructure data are available. This should be highlighted more clearly.

This figure is indeed mostly a qualitative look of the correlation between cross-pol and SMB (Fig. 5 is a quantitive look at the correlation). We agree that the patterns cannot be explained easily and will remove the word "clear" as it is subjective. However we still think that this figure holds value to see where the cross-pol data and the SMB agree with each other and where not. We will highlight the areas where the is disagreement more in the text.

Yes it is unfortunately that DIR and HIR do not have the same datasets available, that of course would be optimal. On the other hand having this many current datasets available for even just one ice rise in Antarctica is already rare.

P12 Figure 6 is also confusing and needs clarification. First, what are the wind

directions (guessing the black lines ?) and how do they represent wind direction? I assume that the P and P' labels mark the start and end of the transects? And the authors should mark all relevant figures including this one, with windward and leeward sides. Also, the axes text is too small.

No the black lines are GPR tracks (However I see this is not mention in the caption. We will add that.).

The wind rose in the corner of the figures shows the wind direction.

P12. Lines 3-8. The authors claim that the density of snow might decrease with a constant addition of new snow, which might be reasonable leading on from Lienss et al 2020 in which the snowpack was located in a forest clearing in Finland where blowing snow is minimal. However, in reality would the windward side of an ice rise not be subjected to the development of a slab layer which would likely result in an increased snow density ? Furthermore, would blowing snow not be more likely to redistribute the snow from the windward to the leeward side of the rise? I understand that these processes are not included in the model/analysis but they are strong controlling factors of a snowpack state when non-flat terrain dominates.

We do not believe that a wind slap would develop on the flank of the ice rise as it is not steep enough (<1%).

Yes, blowing snow is redistributed from the windward side to the leeward side, however redistributed snow is generally denser than fresh snow.

The in-situ SMP measurements show lower densities on the windward side of the ice rise.

P13 Figure 7 and its description on p12-14. Why did the authors simply arithmetically average the microstructure information? A weighted average would be more appropriate given potential variations in each thickness and microstructure. For example, two equally thick layers with very different SSAs will give very different backscatter responses. I would have thought that weighted averages by layer thickness would be far more instructive. Plus it would be instructive to provide the reader with standard deviation of variation of the microstructure. The panel figures are too compressed - more should be made of them to provide better insight into the explanation of the cross-pol ratio data.

We will asses the impact of changing the averaging to a weighted average by layer thickness and provide that as additional results.

EXTRA NOTE:
Actually, upon rereading this, I would like to take my initial answer back. Using a weighted average here would defeat the purpose. I agree that the layer composition can have a large impact on the modelled values. However, here the focus is on the average impact of each parameter (density, grain size, layer

thickness) on their own. If Density was weighted by layer thickness, it could no longer be distinguished from layer-thickness.

P14 Section 4.3. This section is not precise and needs to be written with more clarity. For example, line 14 is not necessarily the case because the averaging of all layer information in Figure 7 masks out the variability of potentially underlying processes that influence the grain radius and/or density values. Simply picking high/low SMB and correlating them with HV/HH and explaining by aggregated grain radius, density is perhaps rather too simplistic.

Figure 8 and 9. What is the difference between the depolarization ratio and the cross-polarization ratio? The authors should be consistent.

This should also be cross-polarization ratio. We will correct that.

P15 Fig. 8
P16 Fig. 9

P14 line 13. Do the authors mean R^2 value or R correlation? Also, for all correlations, the significance level must be included.

Pearson correlation coefficient. We will include significance values for all correlations.

P15 L4

P15 Figure 8C. How are the dotted lines estimated? The authors should explain.

It is simply the Backscatter intensity of the deepest point shown in the figure for both leeward and windward. However I see that this is not mentioned in the figure caption and we will add this.

P15 Fig. 8 Caption

P14 Line 31. Suggest use "vice versa" rather than "the other way around" which is confusing.

We will do that.

P16 L1

P15 Figure 9. Why did the authors choose a 4 point running mean and a 100 point mean for the snow microstructure and cross-pol ratio respectively? 100 pixel

running mean gives an averaging distance of 50x100 = 5km. Why did you not apply the same to the running average to the microstructure data?

A 4-point running mean for the SMP data is also 4x1000 = 4km. As the SMP measurement locations are this far apart. (This is not completely true as they are only 500m apart close to the ice divide). However 4!=5 so we will adjust this to 80 pixels.

P17. Lines 1-12. The question of anisotropy as an explainer is an interesting one. However, two problems emerge. The first is that the authors relate this to fresh snow which could indeed be the case for higher radar frequencies but for C-band, it is unlikely to have an impact at that wavelength - the Lievens et al C-band study (2019) is for deep snow only and is not sensitive to snow less than about 2 m (this is why it is applicable to mountain snow). And the Leinss (2020) study refers to X-Ku band - I would not expect it to be applicable at C-band (S1).

"the Lievens et al C-band study (2019) is for deep snow only and is not sensitive to snow less than about 2 m (this is why it is applicable to mountain snow)."

Could you specify where this is said? We could not find that statement in the paper. On the contrary figure 5 and 7 of the paper show good alignment between their method and in-situ measurements, all with less than 2m of snow height.

P17 lines 13-18. Did the authors experiment by inserting rough layers in the SMRT which I believe is possible ? This might help to formally discount that that possibility.

We did, but the effect was minimal. We will add some more information to this to the discussion.

P17 lines 24-29. This paragraph is confusing as it refers to the windward side only but with contradictory arguments. Also, based on the points above, it is conjecture and inconclusive.

Yes you are correct. In line 28 the "windward" was actually supposed to be a "leeward". This is an oversight and will be corrected.

P18 L18

P17 line 30-P18 line 2. The explanatory discussion can only really come from the analysis of the LIR data since there are no simulations of the other ice rises. This echos the point above about the role of Figure 6 which introduces a tantalizing relationship between SMB and the S1 cross-pol ratio for the HIR data. The only simulation data available are for the LIR for which the explanation is speculative from the analysis. Given that the LIR is the only place to have any explanatory power, this should be made clear at the outset and be clear in the discussion

We will be more clear from the start that our analysis is best on the LIR and HIR and DIR should be only considered auxiliary data points to the analysis.

P3 L11
P7 L3

---

## Author Response (AR2)

**Review of Manuscript 2024-EGUSphere-2024-2077: Sentinel-1 cross-polarization ratio as a proxy for surface mass balance across east Antarctic ice rises**

William David Harcourt, University of Aberdeen September 25, 2024

**Overview**

This study develops an empirical relationship between Surface Mass Balance (SMB) and the cross- polarisation ratio ($\sigma_{HV}/\sigma_{HV}$) of Sentinel-1 SAR data derived from a comparison to in situ snow data acquired from Antarctica. The relationship has been shown to demonstrate clear accumulation patterns across the three ice rises studied. On the windward side, high SMB and cross-polarisation ratio is associated with low density snow and smaller grains, whilst the opposite is true on the leeward side. A key part of the data processing is correcting for the satellite incidence angle which the study computes using a linear regression between SMB and at sampled locations and cross-polarisation ratio. Mapping these patterns shows that this ratio may be used as a proxy for SMB across Antarctica and therefore has potential to help map SMB in areas with few in situ measurements. Because of the potential application of this method, I believe the study should be published subject to the revisions below.

**General Comments**

My general comments can be split into three points:

- I am unsure about how the incidence angle correction has been implemented. Sentinel-1 has an incidence angle file associated with it, which I assume you use (although this is not stated). What is the result if you apply a standard approach such as conversion to $\gamma^0$ or $\sigma^0$ (divide by sin θ) (e.g. Small 2011). As I understand it, the regression coefficients used in Eq. 2 are derived from a regression between the cross-polarization ratio and angle. But you say this is calculated at each sampling point, so it's not clear what data is being used. Rewriting some of the text in Section 3.3 will probably help to clarify these points.

- Each sample point was covered by at least 4 orbits. The incidents angles for each orbit vary (somewhere between 20 and 40 degree (Fig. 3A)). So the

different orbits are whats used to derive the regression. We will rewrite Section 3.3 and try to clarify this.

- P9 L1-2

-     Much of the results section discusses results as if there is a clear pattern between e.g. SMB and cross-polarisation ration. Whilst I can see there is a relationship, to me the pattern is variable and not consistent, implying there is more complex physics at play. Downplaying some of the results and emphasising the variable due to e.g. snowfall variations, local climate might help with this.

- We agree that there are more complex physics at play and the paper is merely trying to look at correlations and discuss possible mechanisms that could drive them. We don't think that we have a solid physical explanation for the observations and we will try to make it more clear that we are not claiming to have one.

-

- P13 L1-2

-     The text is a bit colloquial in places. Phrases such as 'want to' and 'coming from' and 'steady' are used which do not describe some of the underlying processes being discusses e.g. quantifying wind speeds, describing the travel orientation of winds. Editing the text throughout will help here.

- We will rewrite the text to ensure it has a more professional tone, replacing colloquial phrases with more formal language to accurately describe processes.

**Technical Corrections (References to page numbers (P) line (L) num- bers in preprint)**

We will accept all technical corrections and reply to the questions in the following notes.

Abstract

P1L3: 'large spatial coverage and and ability to penetrate the snowpack'

P1 L3

P1L15: You probably want to add that the proxy should be combined with physical models.

P1L15

**Introduction**

P1L19: 'large uncertainties': how large are the uncertainties? Maybe quote mass balance for year e.g. 2023 + uncertainty?

Will do so.

P2 L1-2

P2L6-7: This sentence repeats what you've just said 'as in-situ measurements are sparse'. Suggest remove?

The second sentence is meant specifically towards AWS's, snow stakes and firn cores, whereas the first one is meant generally towards all in-situ measurements. So I don't think its completely redundant.

I Do not think this should be removed? Not sure what to do here?

P2L13: Also different densities of dry snow, wet snow, firn and ice.

P2L15

P2L26: Is this because the increase in travel time due to snow thickness increases is larger for the

co-polarised image (i.e. $\sigma_{VH}$ becomes larger than $\sigma_{VV}$? I think this should be stated clearly.

We will rewrite the sentence to make it more clear.

P2 L29

P2l29: 'ground that ice not ice'?

This comment is unlcear to me,

P2L33: Remove 'want'

P2 L34

P2L35: 'driving the cross-polarization ratio variability, which relates to volume scattering from the snowpack.'

P3 L4

P3L7: Change 'synthetic' to 'theoretical'

P3 L11

Data and study area

P3L17: Not keen on 'island-like topography'; maybe 'a protruding bedrock bump' or something similar?

Personally I would keep 'island-like topography' as a protruding bedrock bump' seems overly complicated.

P3L25: ', this allows for'

P3 L29

P3L26: 'crevassing which creates a strong surface scattering response, the backscatter signal from the undisturbed snowpack will dominate.'

P3 L32

P3L26: Can you comment on the accuracy of ERA5 for interpolating? How well does it match AWS1 data? Also state pixel size.

We will add a sentence for that.

P4 L18-19

P4 L20

P4L15: I would include the tracks on Fig.1 if you can.

Tracks are shown in Fig.1 but hard to see due to zoomed out view.

P4L18-19: State that the dating using ice cores is described below.

P4 L28-29

P4L19: Same GPR system as before? If not, please describe it briefly.

Yes, same system. We will add a sentence noting that.

P4 L26

P4L26: Can you briefly explain of SSA was calculated given that it is included in Eq. 1 below?

We got the SSA form the SMP measurement using the empirical approach explained in Proksch et al. (2015). Is this what you mean, or do you mean what SSA is in general? In this case We can add a short sentence explaining that.

P5 L2, P6 L1-2

Correction: This was correct before I did the changes in blue. The empirical relationship the one by Calonne 2020 which is based on Proksch 2015. Eq. 1 is just a geometric relationship. I reverted the changes and added some more explainations.

P6 L2-4

P6L1: I would include the locations of these samples in Fig. 1.

Would be very hard to see

P6L4-6: Remove repitition (e.g. Sentinel-1). How well distributed were the images across the year? E.e did you have images in particular seasons? Could you also state why you average across 6 years -

my assumption is to remove noise, but snow conditions might change quite significantly from year to year.

We will add an overview of the seasonal distribution of the sentinel-1 data.

P6 L17-18

P6L9: Just to be absolutely clear, I read 'the logarithmic ratio between' as the ratio between $\sigma_{HV}$ and $\sigma_{VV}$ in dB?

Yes.

Data and study area

P6L24: 'to what degree'

P7 L5

P7L3: Is the space between isochrones only related to snow accumulation? What about firn, ice or even ice lenses (maybe not an issue here given the low melt rates)?

This approach is based on the assumption that the snowpack follows the the Herron–Langway firn densification model. So that we can model the density of each layer and use that for the accumulated mass. Therefore ice lenses would indeed be problematic, but as you are saying that is not an issue here due to the low melt rates.

P8L14-19: Combine with paragraph above.

P8L11: I would suggest clearly stating that the linear regression is between cross-polarization ratio and incidence angle (taken from the Sentinel-1 data set).

P9 L2

P8L16: Not sure this makes sense to me 'obscuring the incidence angle correction'

P9L6-8: Not sure I understand how the AWS2 data was extended by 20 years? What does it mean by 'looping'?

The record of the same year (2018) was used repeatedly.

P9 L 22-24

P9L9-P10L2: This section isn't clear to me e.g. how the AWS data was extended.

P10L6: Which panel are you referring to? Also, visually Fig, 4d and e don't align well together, yet you state in the text they do?

Line 6 refers to panel 4C. In the text we wanted to argue that 4a and 4b align well not 4d and e. Since we are talking about grain size there. However upon re-reading the paragraph we agree that this is not clear at all from the text and understand the confusion. We will rewrite the paragraph accordingly.

P9 L31

**Results**

P10L25: 'measured SMB' - important to emphasis what is measured and what is modelled.

P12 L3

P11L1-12: The patterns described in this section are a little vague and I struggle to follow some of it. For example, you state that SMB and cross-polarisation is higher on the windward side, lower on the windward side, but visually this does not match the graphs, they are very variable. Fig. 6A is most clear, so I think you can make the case for this pattern here, but for C and E I would instead on the variability of the pattern. It's still okay to state the broad pattern, but I would refrain from saying it is 1clear'.

We will remove the word "clear".

P12 L11

P11L2: Could you label where the windward side is on the profiles should be for clarity? P13L4: Again, best to annotate windward and leeward side throughout on you figures.

We will do so.

Fig. 9, Fig. 2, Fig. 4

P14L7-13: Similar to above, the patterns are not totally clear. I would suggest being more cautious in your description.

**Discussion**

P16L1-2: This doesn't explain the reason why grain sizes are high on the leeward side & low on the windward side? I would think high SMB would mean higher density due to greater snow compaction? I am possibly misinterpreting things.

Fresh snow has low density and grain size. With time snow compaction will increase the density and grain size. This means that snow near to the surface will have lower density and grain size in areas of high SMB where the snow had less time for compaction.

Unsure what to add here?

P17L33: 'Antarctic'

P18L3-4: What are those 'right circumstances'?

See P17 L32-34: "All three ice rises represent dry snow accumulation zones, which are a result of orographic uplift from steady katabatic winds. Areas with surface melting, or with a very low accumulation rate like the antarctic plateau, might not behave in the same way. "

This was not a change but an excerpt from the discussion. Now these lines can be found on P18 L25 - 27

**Figures**

Figure 2: I find it a little confusing to have a distance profile on top then time series below, can you make 2 separate figures?

We will try to separate them more clearly.

Figure 3: What do the dots represent? Average HV/HH for each pixel?

The linear regression between the variables.

**References**

Small, D. (2011), 'Flattening gamma: Radiometric terrain correction for sar imagery', IEEE Trans- actions on Geoscience and Remote Sensing 49(8), 3081–3093.

P2. Line 16, The authors claim that SAR is sensitive to snow microstructure and is independent of cloud cover. In fact, studies show that cloud cover can impact the radar response and that the SAR the sensitivity to snow microstructure is frequency-dependent. I would encourage the authors to be more precise in their writing.

Thank you for mentioning this. We will include this information.

P1 L19

While I am aware that atmospheric conditions are important for InSAR phase delays I found it hard to find information that shows a non-negligible affect of clouds on C-band backscatter.

P2 line 33. Using the co-pol HH channel will have less sensitivity to volume scattering-dominant processes that are evident in the VV co-pol channel, as used by Lievens et al (2018). Likely this may impact the results so the authors should explain more what the impact might be.

We will add some further explanation on that.

P2 L35
P3 L1-2

P3 Line 15. The locations of the LIR, HIR and DIR are not labelled in Fig 1 making this quite difficult to assess and understand what leeward/windward actually means.

Yes I agree, the ice rises should be labeled in Fig. 1. We will add labels.

P5 Fig. 1

P4 Line 11. Since ERA5 was used to gap fill in 2018, the authors should provide an assessment of the uncertainty of this gap-filling since wind speed has a significant impact on the SNOWPACK estimates.

We are confident in the ERA5 wind data as it is in good agreement with the AWS wind speeds in 2019 (Figure 2). However we are open to the idea of including a quantitive uncertainty assessment of the ERA5 data.

P4 L18-19

P4 eq 1. Please define all units used.

We will do so.

P5 L4 Eq. 1

Unit for GS added

P4 Line 27, Equation (1) and P5 lines 1-5 and throughout the paper. The authors should be more specific in their language referring to "grain size". This variable is a critical parameter in the SMRT estimation process and there are several emerging terms regarding what is meant by grain size (effective grain size, measured or observed grain size, optical grain size). Also, the concept of a "real grain size" is somewhat misleading.

We agree that "grain size" is an imprecise name. However further along in the text it is explained what exactly is meant by that (P5,L2-3). So I think the best nomenclature I could think of would be "estimated grain size"?

EXTRA NOTE:
Changed to optically equivalent grain diameter throughout the manuscript

Double checked should be replaced everywhere now.

P5 Fig 1. Does the scale bar for B (HH pol) apply to the cross-pol in C? This should be stated. Also, the GPR tracks are not clear - the authors provide a more detailed map of these tracks at the 3 locations.

Yes the scale bar does apply to Fig. 1C as well, we will make that more clear.

The tracks shown in the figure are the same tracks as shown later. The problem is that this is a more zoomed out view making them hard to see. I think form a graphic design point of view it will unfortunately be difficult to have them clearly visible on this zoom level without dramatically increasing the size of the figure. However we will try to improve this as much as possible.

P5 Fig. 1 Caption

The tracks are shown in detail in Fig. 6

P6 lines 14-15. 50 m spatial resolution is much finer resolution than the Lievens (2019) approach. They have noted in their paper that the smoothed backscatter data was posted to 1 km. Why did you select 50 m?

We chose a higher spatial resolution for the Sentinel 1 data since we also have higher resolution data available from the GPR for the correlation. This allowed us o maximise the number of data points available for analysis.

P6 Section 2.5. Did the authors include speckle filtering in their workflow? Even for EW data, speckle noise may have an impact and when the data are averaged, the speckle (multiplication noise) could have an impact on the averaging process of the S1 data. How do they know that this doe not have an effect?

No speckle filtering was included. We will analyse the impact of a speckle filter on the results.

P8 Fig 3
P11 Fig. 5
P9 L11-12

P9 Line 7. Can the authors explain what they mean by "looping the 2018 input AWS for 20 years"?

The record of the same year (2018) was used repeatedly.

P9 L25 -29

P10 Line 3-6. I disagree that there is "good agreement between SNOWPACK grain radius (?) and the SMP snow grain radius (?). There is much more variability in the SMP data than observed by the model indicating a lack of model sensitivity. Can the authors explain what this might be caused by and the importance of this?

We agree that the term "good agreement" is subjective and we will remove it. It is however expected that SNOWPACK does not fully capture small-scale variability observed in the SMP data. We will clarify this in the manuscript by noting that such variability is not represented in the model.

P9 L31-31

P10 L9-10

P10 Line 21. Penetration depth in microwave research is defined as 1/e. Is this what the authors mean or do they mean the maximum depth beyond which no further response is observed?

The maximum depth beyond which no further response is observed.

EXTRA NOTE:
Changed penetration depth to effective penetration depth throughout the manuscript

Should now be changed everywhere

P10 Section 4.1. I know this is pedantic but the authors seem to conflate Correlation R with coefficient of determination ($R^2$) which is the measure of the fit of a linear regression. Perhaps they can be consistent in their use of such standard terms.

Figure 5 should include a legend of the colours for improved clarity.

We will add a colour legend.

P11 Fig. 5

P10 Section 3.5. The authors state that they use a stickiness value of 0.15 for all runs. How was this value selected and how sensitive are the results to it?

Generally speaking both hh and hv decrease with increasing stickiness, with hv decreasing faster, resulting in a generally lower hv/hh ratio with increasing stickiness. However, since the same stickiness is used everywehere, it does not introduce any relative variability along track and therefore has little impact on the correlation. Nether the less we will add some more background on this to the method part.

P11 L1-4

P11 L15

P10 Section 3.5. Why did the authors select the IBA and not, for example the DMRT approach. It would be helpful for the reader to provide this justification. Furthermore, what was the substrate condition used in the model - was it an infinite background somehow? A more comprehensive explanation of the model set-up would certainly help the reader follow the logic here.

We will asses the impact different settings of the radiative transfer model on the results.

P11 L8 - 10

P10 The authors should include standard error metrics of the regression lines (the slope coefficient). What is the variability of the regression coefficients calculated? And how is this calculated?

We will include  standard error metrics.

P11 Fig. 5 Caption

P11/12. The role of Figure 6 is unclear. I understand it shows the SMB variations with cross-pol ratio but the patterns cannot be explained easily, despite the authors asserting that correspondence between SMD and cross-pol ratio is "clear". I can see that there is correspondence between the SMB and the cross-pol ratio for the HIR but for the LIR it is somewhat related but the DIR has only a moderate correspondence. It is unfortunate that in situ data are not available for the DIR and especially the HIR location where there is indeed the strongest agreement. The authors conduct an analysis of LIR based on the SMP, SNOWPACK, SMRT and cross-pol data. But no similar analysis can be undertaken of DIR and HIR because no microstructure data are available. This should be highlighted more clearly.

This figure is indeed mostly a qualitative look of the correlation between cross-pol and SMB (Fig. 5 is a quantitive look at the correlation). We agree that the patterns cannot be explained easily and will remove the word "clear" as it is subjective. However we still think that this figure holds value to see where the cross-pol data and the SMB agree with each other and where not. We will highlight the areas where the is disagreement more in the text.

Yes it is unfortunately that DIR and HIR do not have the same datasets available, that of course would be optimal. On the other hand having this many current datasets available for even just one ice rise in Antarctica is already rare.

P12 Figure 6 is also confusing and needs clarification. First, what are the wind directions (guessing the black lines ?) and how do they represent wind direction? I assume that the P and P' labels mark the start and end of the transects? And the authors should mark all relevant figures including this one, with windward and leeward sides. Also, the axes text is too small.

No the black lines are GPR tracks (However I see this is not mention in the caption. We will add that.).

The wind rose in the corner of the figures shows the wind direction.

P12. Lines 3-8. The authors claim that the density of snow might decrease with a constant addition of new snow, which might be reasonable leading on from Lienss et al 2020 in which the snowpack was located in a forest clearing in Finland where blowing snow is minimal. However, in reality would the windward side of an ice rise not be subjected to the development of a slab layer which would likely result in an increased snow density ? Furthermore, would blowing snow not be more likely to redistribute the snow from the windward to the leeward side of the rise? I understand that these processes are not included in the model/analysis but they are strong controlling factors of a snowpack state when non-flat terrain dominates.

We do not believe that a wind slap would develop on the flank of the ice rise as it is not steep enough (<1%).

Yes, blowing snow is redistributed from the windward side to the leeward side, however redistributed snow is generally denser than fresh snow.

The in-situ SMP measurements show lower densities on the windward side of the ice rise.

P14 L1-2

P13 Figure 7 and its description on p12-14. Why did the authors simply arithmetically average the microstructure information? A weighted average would be more appropriate given potential variations in each thickness and microstructure. For example, two equally thick layers with very different SSAs will give very different backscatter responses. I would have thought that weighted averages by layer thickness would be far more instructive. Plus it would be instructive to provide the reader with standard deviation of variation of the microstructure. The panel figures are too compressed - more should be made of them to provide better insight into the explanation of the cross-pol ratio data.

We will asses the impact of changing the averaging to a weighted average by layer thickness and provide that as additional results.

EXTRA NOTE:
Actually, upon rereading this, I would like to take my initial answer back. Using a weighted average here would defeat the purpose. I agree that the layer composition can have a large impact on the modelled values. However, here the focus is on the average impact of each parameter (density, grain size, layer thickness) on their own. If Density was weighted by layer thickness, it could no longer be distinguished from layer-thickness.

Increased size of Fig. 7

Regarding the editor Note:
I think there might be a misunderstanding. What is shown in the figure is the average density of the first 15 meters of depth. Not the first 15 layers. It already gives a relationship with mass?

Otherwise I might have misunderstood something here.

P14 Section 4.3. This section is not precise and needs to be written with more clarity. For example, line 14 is not necessarily the case because the averaging of all layer information in Figure 7 masks out the variability of potentially underlying processes that influence the grain radius and/or density values. Simply picking high/low SMB and correlating them with HV/HH and explaining by aggregated grain radius, density is perhaps rather too simplistic.

I agree that the approach is simplistic, but that is whole point of it. It looks at empirical relationships that might be useful.

Figure 8 and 9. What is the difference between the depolarization ratio and the cross-polarization ratio? The authors should be consistent.

This should also be cross-polarization ratio. We will correct that.

P15 Fig. 8
P16 Fig. 9

P14 line 13. Do the authors mean R^2 value or R correlation? Also, for all correlations, the significance level must be included.

Pearson correlation coefficient. We will include significance values for all correlations.

P15 L4
P15 L13

P15 Figure 8C. How are the dotted lines estimated? The authors should explain.

It is simply the Backscatter intensity of the deepest point shown in the figure  for both leeward and windward. However I see that this is not mentioned in the figure caption and we will add this.

P15 Fig. 8 Caption

P14 Line 31. Suggest use "vice versa" rather than "the other way around" which is confusing.

We will do that.

P16 L1

P15 Figure 9. Why did the authors choose a 4 point running mean and a 100 point mean for the snow microstructure and cross-pol ratio respectively? 100 pixel running mean gives an averaging distance of 50x100 = 5km. Why did you not apply the same to the running average to the microstructure data?

A 4-point running mean for the SMP data is also 4x1000 = 4km. As the SMP measurement locations are this far apart. (This is not completely true as they are only 500m apart close to the ice divide). However 4!=5 so we will adjust this to 80 pixels.

There was a mistake. The running mean in Fig. 9 was 10 pixels not 100. However I now changed it to 80.

P16 Fig. 9, Fig. 9 caption

P17. Lines 1-12. The question of anisotropy as an explainer is an interesting one. However, two problems emerge. The first is that the authors relate this to fresh snow which could indeed be the case for higher radar frequencies but for C-band, it is unlikely to have an impact at that wavelength - the Lievens et al C-band study (2019) is for deep snow only and is not sensitive to snow less than about 2 m (this is why it is applicable to mountain snow). And the Leinss (2020) study refers to X-Ku band - I would not expect it to be applicable at C-band (S1).

"the Lievens et al C-band study (2019) is for deep snow only and is not sensitive to snow less than about 2 m (this is why it is applicable to mountain snow)."

Could you specify where this is said? We could not find that statement in the paper. On the contrary figure 5 and 7 of the paper show good alignment between their method and in-situ measurements, all with less than 2m of snow height.

P17 lines 13-18. Did the authors experiment by inserting rough layers in the SMRT which I believe is possible ? This might help to formally discount that that possibility.

We did, but the effect was minimal. We will add some more information to this to the discussion.

Here my answer was not very precise, we did experiment with inserting INTERNAL layers into the snow pack. However modelling surface scattering from a rough surface is not currently possible in SMRT, at least to my knowledge. I left this out, since we talk about surface scattering in the paragraph.

P17 lines 24-29. This paragraph is confusing as it refers to the windward side only but with contradictory arguments. Also, based on the points above, it is conjecture and inconclusive.

Yes you are correct. In line 28 the "windward" was actually supposed to be a "leeward". This is an oversight and will be corrected.

P18 L18

P17 line 30-P18 line 2. The explanatory discussion can only really come from the analysis of the LIR data since there are no simulations of the other ice rises. This echos the point above about the role of Figure 6 which introduces a tantalizing relationship between SMB and the S1 cross-pol ratio for the HIR data. The only simulation data available are for the LIR for which the explanation is speculative

from the analysis. Given that the LIR is the only place to have any explanatory power, this should be made clear at the outset and be clear in the discussion

We will be more clear from the start that our analysis is best on the LIR and HIR and DIR should be only considered auxiliary data points to the analysis.

P3 L11
P7 L3

---

## Editor Decision (ED2)

Kausch et al 16th Sept 2025

After a careful revisit of the author's responses to the reviewer comments, there are a number of areas that have not been addressed. There are also additional clarifications and modifications needed (with apologies that I did not pick these up the first time around).

In response to the reviewer comments:
'*We agree that there are more complex physics at play and the paper is merely trying to look at correlations and discuss possible mechanisms that could drive them. We don't think that we have a solid physical explanation for the observations and we will try to make it more clear that we are not claiming to have one.'*

I am sorry to say I still cannot find this in the text at P13 L1-2, either in the tracked change or actual version.

'*I Do not think this should be removed? Not sure what to do here?'*

As I have already communicated to you, I agree with the reviewer and already asked for this change. Including this additional sentence, while makes sense to you, makes less sense to the reader. I advise against retaining it, but as you feel so strongly about this I will not insist on this change.

'*P4L15: I would include the tracks on Fig.1 if you can.*
*Tracks are shown in Fig.1 but hard to see due to zoomed out view.'*

Please could you make the tracks thicker?

*P8L14-19: Combine with paragraph above.*
This has not been done.

*P8L16: Not sure this makes sense to me 'obscuring the incidence angle correction'*
No change has been made to the text – please clarify

*P9L9-P10L2: This section isn't clear to me e.g. how the AWS data was extended.*
Please could you respond to this point

*P16L1-2: This doesn't explain the reason why grain sizes are high on the leeward side & low on the windward side? I would think high SMB would*

*mean higher density due to greater snow compaction? I am possibly misinterpreting things.*

*Fresh snow has low density and grain size. With time snow compaction will increase the density and grain size. This means that snow near to the surface will have lower density and grain size in areas of high SMB where the snow had less time for compaction.*

*Unsure what to add here?*

Please add text to the document, but this explains surface discrepancies, not lower in the pack where you might expect higher densities with high SMB. What is the reason for the low density with high SMB? This ties in to a general discussion on whether the snowpack model is representing density and microstructure lower in the pack. If not, the radiative transfer model may not be able to represent the backscatter in a manner that reflects observations.

*Figure 2: I find it a little confusing to have a distance profile on top then time series below, can you make 2 separate figures?*

*We will try to separate them more clearly.*

This has not been done

*Figure 3: What do the dots represent? Average HV/HH for each pixel?*

*The linear regression between the variables.*

I think the reviewer is asking about the processing of data here

*While I am aware that atmospheric conditions are important for InSAR phase delays I found it hard to find information that shows a non-negligible affect of clouds on C-band backscatter*

Please mention this may be source of error and cite the InSAR studies

*No speckle filtering was included. We will analyse the impact of a speckle filter on the results*

Speckle is only mentioned twice in the document. I see it has now been included, but no analysis of its effect has been included

*However modelling surface scattering from a rough*
*surface is not currently possible in SMRT, at least to my knowledge*

(just a comment – yes this is possible)
* * *
Additional corrections

Page 14 line 4. SSA is not a measurement of grain shape – there is no way to recover grain shape from SSA. There are some correlations i.e. fresh crystals generally have high SSA, depth hoar have low SSA, but it's the tail of the correlation function that has more information on the shape. SSA is only related to the correlation function at the origin. See https://doi.org/10.1029/2021AV000630. This ties in with Pg 6, line 7.

It is slightly misleading to say high SSA is a sign of a non-spherical grain shape. In any case, high SSA crystals do not scatter much so these are not so important. See also page 17, line 7.

Page 13, line 5. SSA is not exclusive to SMP: SSA is mathematically equivalent to optical equivalent grain diameter through equation 1.

Page 17, line 2 / figure 9B. I'm not comfortable with the reliance on the SMP relationship with cross-pol as the SMP only extends to the top 1m, and doesn't consider SSA lower in the snowpack where the bulk of the scattering is occurring

Page 17, line 4. 'Therefore we argue…' I'm afraid I simply don't understand this statement. Please could you elaborate?

Page 18, line 2. Radar sensitivity to grain extent: this does not take multiple scattering into account

Page 19, line 9. Fresh snowfall may be anisotropic, but this may quickly evolve. This is not a strong argument.

---

## Author Response (AR3)

Kausch et al 16th Sept 2025

After a careful revisit of the author's responses to the reviewer comments, there are a number of areas that have not been addressed. There are also additional clarifications and modifications needed (with apologies that I did not pick these up the first time around).

*Thank you for your additional remarks. We have now revised the manuscript to address all remaining issues. Please find the latest answers below in green. We also numbered the responses here for a better overview. All page and line numbers refer to the track changes version of the manuscript.*

In response to the reviewer comments:

'*We agree that there are more complex physics at play and the paper is merely trying to look at correlations and discuss possible mechanisms that could drive them. We don't think that we have a solid physical explanation for the observations and we will try to make it more clear that we are not claiming to have one.*'

I am sorry to say I still cannot find this in the text at P13 L1-2, either in the tracked change or actual version.

*1. It was moved to the previous page by later changes. My bad, it is now at P13 L8-9.*

'*I Do not think this should be removed? Not sure what to do here?*'

As I have already communicated to you, I agree with the reviewer and already asked for this change. Including this additional sentence, while makes sense to you, makes less sense to the reader. I advise against retaining it, but as you feel so strongly about this I will not insist on this change.

*2. Removed the sentence.*

'*P4L15: I would include the tracks on Fig.1 if you can.* *Tracks are shown in Fig.1 but hard to see due to zoomed out view.*'

Please could you make the tracks thicker?

*3. Doubled line thickness. P5*

*P8L14-19: Combine with paragraph above.*

This has not been done.

*4. Done now. (For some reason this is not visible in the track_changes version but only the new manuscript.) P9 L7*

*P8L16: Not sure this makes sense to me 'obscuring the incidence angle correction'*
No change has been made to the text – please clarify

*5. Rephrased the sentence. The point I am trying to make is that by applying it per pixel, spatial variability does no longer play a role. P9 L10*

*P9L9-P10L2: This section isn't clear to me e.g. how the AWS data was extended.*
Please could you respond to this point

*6. Rewrote the paragraph. P10 L7 - P11 L2*

*P16L1-2: This doesn't explain the reason why grain sizes are high on the leeward side & low on the windward side? I would think high SMB would*

*mean higher density due to greater snow compaction? I am possibly misinterpreting things.*

*Fresh snow has low density and grain size. With time snow compaction will increase the density and grain size. This means that snow near to the surface will have lower density and grain size in areas of high SMB where the snow had less time for compaction.*

*Unsure what to add here?*

Please add text to the document, but this explains surface discrepancies, not lower in the pack where you might expect higher densities with high SMB. What is the reason for the low density with high SMB? This ties in to a general discussion on whether the snowpack model is representing density and microstructure lower in the pack. If not, the radiative transfer model may not be able to represent the backscatter in a manner that reflects observations.

*7. Added some explanation. P17 L1-8*

*Figure 2: I find it a little confusing to have a distance profile on top then time series below, can you make 2 separate figures?*
*We will try to separate them more clearly.*
This has not been done

*8. Separated into 2 figures. P7*

*Figure 3: What do the dots represent? Average HV/HH for each pixel?*

*The linear regression between the variables.*

I think the reviewer is asking about the processing of data here

*9. Yes average HV/HH. Added that its the average there. P9, P12*

*While I am aware that atmospheric conditions are important for InSAR phase delays I found it hard to find information that shows a non-negligible affect of clouds on C-band backscatter*
Please mention this may be source of error and cite the InSAR studies

*10. I'm afraid I can not follow. The InSAR studies can not be used as a source to claim that atmospheric conditions may be a source of error, as they only claim this is a problem for InSAR which is a different method and not used in this study.*

*No speckle filtering was included. We will analyse the impact of a speckle filter on the results*
Speckle is only mentioned twice in the document. I see it has now been included, but no analysis of its effect has been included

*11. Added in sentence in results that there was a modest but insignificant improvement in correlation due to adding a speckle filter. P12 L6-7*

*However modelling surface scattering from a rough surface is not currently possible in SMRT, at least to my knowledge*
(just a comment – yes this is possible)

Additional corrections
Page 14 line 4. SSA is not a measurement of grain shape – there is no way to recover grain shape from SSA. There are some correlations i.e. fresh crystals generally have high SSA, depth hoar have low SSA, but it's the tail of the correlation function that has more information on the shape. SSA is only related to the correlation function at the origin. See https://doi.org/10.1029/2021AV000630. This ties in with Pg 6, line 7.

*12. Rephrased the sentence and clarified that SSA is not a direct measurement of grain shape, but associated with it. P15 L5-7*

It is slightly misleading to say high SSA is a sign of a non-spherical grain shape. In any case, high SSA crystals do not scatter much so these are not so important. See also page 17, line 7.

*13. Would a sphere not always have the lowest SSA geometrically possible?*

Page 13, line 5. SSA is not exclusive to SMP: SSA is mathematically equivalent to optical equivalent grain diameter through equation 1.

*14. Sorry, what was meant here is that within the dataset used in this study it is exclusive. It is of course not exclusive to SMP measurements in general. This however, was poorly phrased and I added that info. P14 L1*

Page 17, line 2 / figure 9B. I'm not comfortable with the reliance on the SMP relationship with cross-pol as the SMP only extends to the top 1m, and doesn't consider SSA lower in the snowpack where the bulk of the scattering is occurring

*15. This is a shortcoming indeed. I added an extra sentence mentioning this. P18 L7-8*

Page 17, line 4. 'Therefore we argue...' I'm afraid I simply don't understand this statement. Please could you elaborate?

*16. Rephrased the whole paragraph. P17 L9 - P18 L8*

Page 18, line 2. Radar sensitivity to grain extent: this does not take multiple scattering into account

*17. Added a line that multiple scattering is not considered in this scenario. P19 L7*

Page 19, line 9. Fresh snowfall may be anisotropic, but this may quickly evolve. This is not a strong argument.

*18. I agree that this is a shortcoming of this argument and added a sentence to mention this. P19 L25-26*

---

## Author Response (AR4)

**DEAR MELODY SANDELLS,**

Thank your very much for your reviews. We agree with your proposed changes and implemented all corrections as you recommended. We decided to keep figure 1 as it is.

Thanks again and best wishes to you too,
Thore Kausch

**LIST OF CHANGES:**

P9 L2-3
P12 L6
P15 l4-8
P17 L4-8